# A room-temperature sodium–sulfur battery with high capacity and stable cycling performance

Xiaofu Xu[1,2], Dong Zhou[3], Xianying Qin[1,2], Kui Lin[1,2], Feiyu Kang[1,2],
Baohua Li[1,2], Devaraj Shanmukaraj[4], Teofilo Rojo[4], Michel Armand[4] & Guoxiu Wang [3]

High-temperature sodium–sulfur batteries operating at 300–350 °C have been commercially applied for large-scale energy storage and conversion. However, the safety concerns greatly inhibit their widespread adoption. Herein, we report a room-temperature sodium–sulfur battery with high electrochemical performances and enhanced safety by employing a "cocktail optimized" electrolyte system, containing propylene carbonate and fluoroethylene carbonate as co-solvents, highly concentrated sodium salt, and indium triiodide as an additive. As verified by first-principle calculation and experimental characterization, the fluoroethylene carbonate solvent and high salt concentration not only dramatically reduce the solubility of sodium polysulfides, but also construct a robust solid-electrolyte interface on the sodium anode upon cycling. Indium triiodide as redox mediator simultaneously increases the kinetic transformation of sodium sulfide on the cathode and forms a passivating indium layer on the anode to prevent it from polysulfide corrosion. The as-developed sodium–sulfur batteries deliver high capacity and long cycling stability.

[1] Graduate School at Shenzhen, Tsinghua University, Shenzhen 518055, China. [2] School of Materials Science and Engineering, Tsinghua University, Beijing 100084, China. [3] School of Mathematical and Physical Sciences, University of Technology Sydney, Sydney, NSW 2007, Australia. [4] CIC ENERGIGUNE, Parque Tecnológico de Álava, Miñano 01510, Spain. These authors contributed equally: Xiaofu Xu, Dong Zhou. Correspondence and requests for materials should be addressed to B.L. (email: libh@mail.sz.tsinghua.edu.cn) or to M.A. (email: marmand@cicenergigune.com) or to G.W. (email: Guoxiu.Wang@uts.edu.au)

To date, batteries based on alkali metal-ion intercalating cathode and anode materials, such as lithium-ion batteries, have been widely used in modern society from portable electronics to electric vehicles[1]. However, batteries based on such intercalation chemistry can only deliver limited energy density, which cannot meet the growing demand for large-scale energy storage[2]. Consequently, alkali metal-sulfur batteries based on a conversion chemistry have attracted tremendous attention due to the high-energy density[3], non-toxicity, and low cost of sulfur (S)[4]. From sustainability and economic points of view, sodium (Na) is a better option than lithium (Li) to couple with sulfur cathode, because of the analogous chemical properties but much higher natural abundance of Na compared with Li[5,6]. Traditional high-temperature Na–S batteries were first commercialized in utility-scale stationary power applications in 2002, based on the following reaction[7]:

$$2Na + nS \leftrightarrow Na_2S_n (n \geq 3) \quad (1)$$

This rechargeable battery system has significant advantages of high theoretical energy density (760 Wh kg$^{-1}$, based on the total mass of sulfur and Na), high efficiency (~100%), excellent cycling life and low cost of electrode materials, which make it an ideal choice for stationary energy storage[8,9]. However, the operating temperature of this system is generally as high as 300–350 °C to ensure a sufficient conductivity of sodium β-alumina solid-electrolyte and keep the polysulfides in a molten state, far exceeding the melting points of Na (98 °C) and sulfur (115 °C)[10]. Such high temperature not only increases the cost of operation and maintenance, but also brings serious safety hazard due to the highly active molten electrodes, which directly restraints the widespread applications of high-temperature Na–S batteries[5,8]. As a result, great efforts have been devoted to lower the working temperature and develop room-temperature Na–S batteries with enhanced safety.

Room-temperature Na–S batteries have been reported since 2006[11]. They have an increased energy density (1274 Wh kg$^{-1}$) compared with high-temperature Na–S batteries[12], because Na$_2$S instead of Na polysulfides is the final discharge product[13,14]:

$$2Na + pS \leftrightarrow Na_2S_p (p \geq 1) \quad (2)$$

However, room-temperature Na–S batteries generally suffered from low reversible capacity, self-discharging, and serious cycling problems. This is mainly due to the poor compatibility between electrodes and electrolyte[12,15]. As for the sulfur cathode, Na polysulfides formed as intermediates during the charge/discharge processes are highly soluble in liquid electrolytes. They can easily shuttle to the Na anode and undergo redox reactions to form lower-order polysulfides, depositing on the Na anode and leading to the loss of active materials and an interfacial deterioration. This greatly decreases the cycling stability of Na–S batteries[16]. Moreover, the cathodic reaction from sulfur to Na$_2$S always accompanies huge volumetric changes, which can easily cause active material shedding[14]. Meanwhile, poor kinetics of the transition from short-chain Na polysulfides or Na$_2$S to long-chain polysulfides leads to low Coulombic efficiency[17]. As for the anode, the highly reactive Na metal can react with most organic electrolyte solvents and Na salts, and then forms a solid-electrolyte interface (SEI) layer featuring ionic conduction but electronic insulation[18,19]. The strength of the SEI layer generally cannot bear the mechanical deformation during Na-ion plating/stripping processes, and hence leads to formation of surface defects and then growth of dendrites from these defects[20]. Such Na dendrites can pierce through the separator, and cause serious safety problems such as short circuits with thermal runaway[21]. They also result in a continuous damage/regeneration of the SEI upon prolonged cycling, which significantly decreases the Coulombic efficiency of batteries[22]. Furthermore, the polysulfides can diffuse through the SEI layer and corrode the Na anode, which causes irreversible capacity loss[23]. All these drawbacks have severely hindered the development of room-temperature Na–S batteries.

Many efforts have been devoted to overcome the above problems, including infusing sulfur into a conductive matrix[16,24,25], surface coatings on sulfur or introducing additives in cathode composition[26], applying sulfides as cathode materials[17,27], employing functionalized separators or interlayers[28–30], and optimizing the electrolyte components. Among them, employing innovative electrolytes have been proposed as one of the most promising strategies to address the inherent drawbacks of room-temperature Na–S batteries without sacrificing the energy density or introducing tedious preparation process[8,15,31]. However, to date, there has still been a lack of fundamental breakthroughs in electrolyte development, which can provide a satisfactory energy density with stable long-term cycling.

Herein, we provide fundamental scientific study on the electrochemical properties of carbonate-based electrolyte in room-temperature Na–S batteries, and report a multifunctional carbonate-based electrolyte consisting of propylene carbonate (PC) and fluoroethylene carbonate (FEC) as co-solvents, highly concentrated bis(trifluoromethane)sulfonimide sodium (NaTFSI) salt, and indium triiodide (InI$_3$) additive. The solubility of Na polysulfides has been efficiently suppressed via the high salt concentration and FEC-rich solvent. Meanwhile, the Na metal anode is effectively protected by an indium (In) layer formed from an In$^{3+}$ redox reaction and a fluorine (F)-rich stable SEI film. In addition, the irreversible Na$_2$S formed during the charging process is remarkably oxidized by I$_3^-$ from the InI$_3$ additive, which greatly increases the Coulombic efficiency. The electrode/electrolyte interfacial phenomena were analyzed by experimental characterizations and theoretical calculations. The as-developed Na–S batteries exhibited outstanding performances with a specific capacity of 1170 mAh g$^{-1}$ (based on the mass of sulfur) at 0.1 C and an extended cycle life.

## Results

**Design and characterization of the electrolytes.** In this study, multiporous carbon fibers (MPCFs) with a large Brunauer-Emmett-Teller (BET) surface area of 2475 m$^2$ g$^{-1}$ were synthesized as the matrix material for sulfur storage (the corresponding synthesis route and characterization of MPCFs are shown in Supplementary Figs. 1 and 2). S@MPCF electrodes were fabricated using sodium carboxymethyl cellulose (CMCNa) as the binder (which can form a strong S–O bond with sulfur to greatly enhance the reversible capacity of Na–S batteries; details are shown in Supplementary Fig. 3). Considering room-temperature Na–S batteries with ether-based electrolytes (such as tetraethylene glycoldimethyl ether (TEGDME) and a combination of 1,3-dioxolane/1,2-dimethoxyethane (DOL/DME)) usually suffer from limited capacity and cycle life[28,32], PC (a representative carbonate ester commonly used in sodium-ion batteries[33]) is coupled with NaTFSI salts as the baseline electrolyte (details are shown in Supplementary Figs. 4 and 5). It should be noticed that in Li–S batteries, the nucleophilic sulfide anions actively react with carbonate solvents via nucleophilic addition or substitution reaction, which results in a rapid capacity fading[34]. However, the side reactions between Na polysulfides and carbonate solvents are much less severe than those between Li polysulfides and carbonate solvents (Supplementary Fig. 6). This may be due to the fact that the larger ionic radius of Na$^+$ than Li$^+$ leads to less dissociation in polar solvents, which results in a lower reactivity of

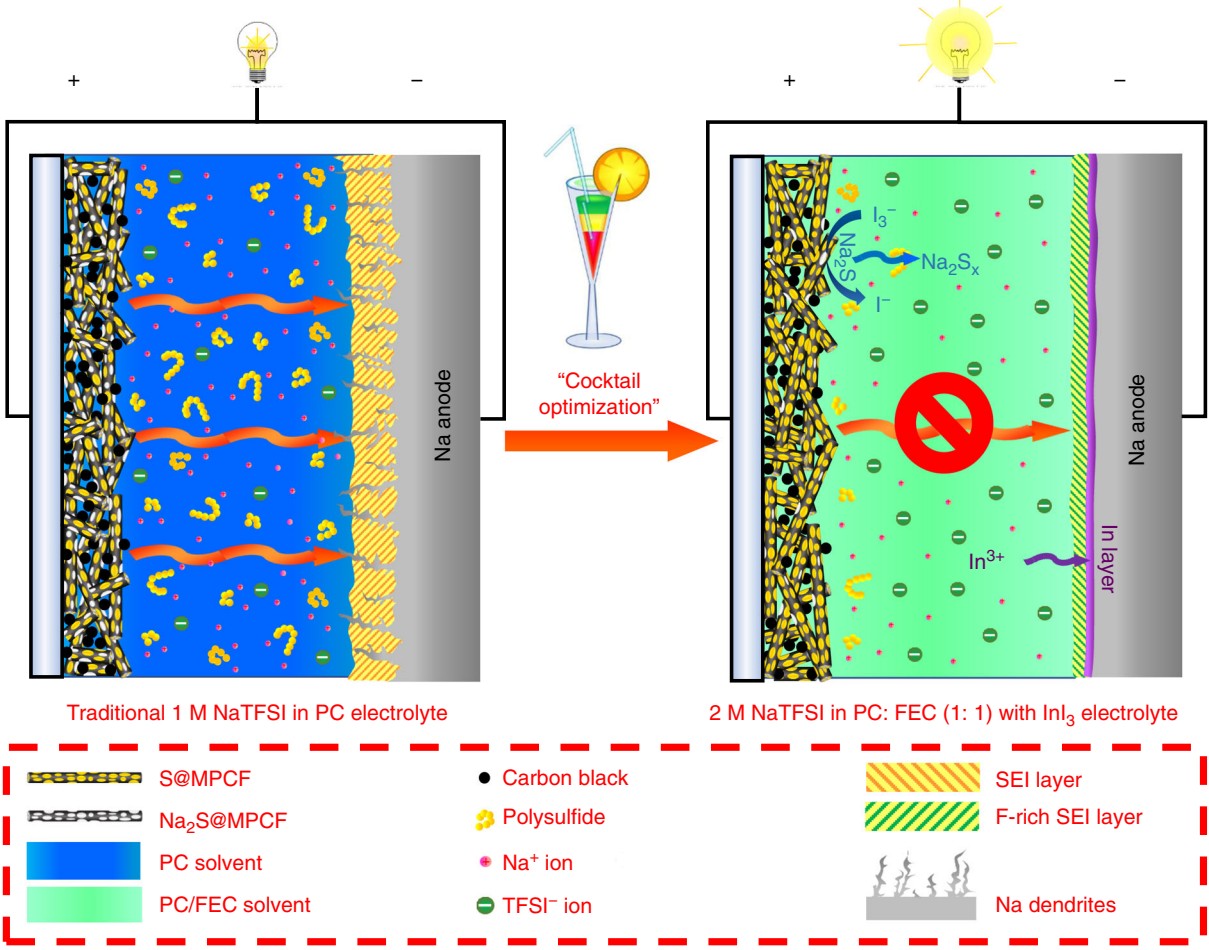

**Fig. 1** Schematic illustration of room-temperature Na–S batteries using (left) conventional 1 M NaTFSI in PC electrolyte and (right) 2 M NaTFSI in PC: FEC (1:1 by volume) with 10 mM InI₃ additive electrolyte

Na⁺-polysulfide⁻ ion pairs than that of Li⁺-polysulfide⁻ ion pairs[35,36]. Therefore, the negative effect of such side reactions on the electrochemical performance of Na–S batteries is negligible.

To further improve the limited capacity (~400 mAh g⁻¹) and cycle life of 1 M NaTFSI in PC electrolyte-based Na–S cells, we developed a novel carbonate-based electrolyte. Figure 1 shows a schematic illustration of the optimization mechanism for this electrolyte in room-temperature Na–S batteries. In conventional PC-based electrolyte, it is widely believed that the transitions from solid-state short-chain polysulfides or Na₂S to long-chain polysulfides are kinetically difficult due to dramatic volume change caused by the large ionic size of Na⁺[37,38]. The nonconductive unconverted Na₂S usually accumulates on the cathode, reducing the charge transfer rate and blocking the ion accessibility, which results in serious polarization as well as gradual capacity fading[38]. Moreover, the shuttle of highly soluble Na polysulfides not only causes the loss of active materials in the cathode, but also leads to the formation of a SEI layer with high Ohmic resistance. Additionally, the growth of Na dendrites also results in low Coulombic efficiency, which is associated with greater safety hazards (Fig. 1, left). Conversely, in our new electrolyte composed of highly concentrated NaTFSI salt and InI₃ additive dissolved in PC/FEC (1:1 by volume) co-solvents, the FEC solvent and high salt concentration not only significantly decrease the solubility of Na polysulfides, but also form a stable F-rich SEI and a dendrite-free Na surface during cycling. The In³⁺ ions from the InI₃ additive construct a passivating In layer on the

anode, which protects against polysulfide corrosion. Meanwhile, the iodide (I⁻) ions can be reversibly oxidized into triiodide ions (I₃⁻) in the charging process[39], facilitating the transformation of Na₂S to Na polysulfides and, therefore, lowering the high irreversibility of Na₂S during the charge/discharge process (Fig. 1, right). Such "cocktail optimization" of electrolyte is expected to ensure an excellent cycling stability for both cathodes and anodes in room-temperature Na–S batteries.

Figure 2a shows the ionic conductivities of 1 M NaTFSI in PC, 1 M NaTFSI in PC: FEC (1:1 by volume), 2 M NaTFSI in PC: FEC (1:1 by volume) and 2 M NaTFSI in PC: FEC (1:1 by volume) with 10 mM InI₃. The plots of log σ vs. T⁻¹ for all the electrolyte samples exhibits a non-linear relationship, which is well fitted by the Vogel-Tamman-Fulcher (VTF) empirical equation below[40]:

$$\sigma = \sigma_o T^{-1/2} \exp\left(-\frac{E_a}{R(T - T_o)}\right) \quad (3)$$

where $E_a$ is the pseudo-activation energy, $\sigma_o$ is the pre-exponential factor, $T_o$ is the ideal glass transition temperature, and $R$ is the gas constant. The fitting parameters and ionic conductivity values are listed in Supplementary Table 2. It can be seen that the ionic conductivities of electrolytes slightly decrease with increasing Na salt concentration and FEC proportion due to the rise in viscosity[41] (Supplementary Fig. 7), and are almost unchanged after the addition of the small amount of InI₃ additive. The 2 M NaTFSI in PC: FEC (1:1 by volume) with InI₃ electrolyte can

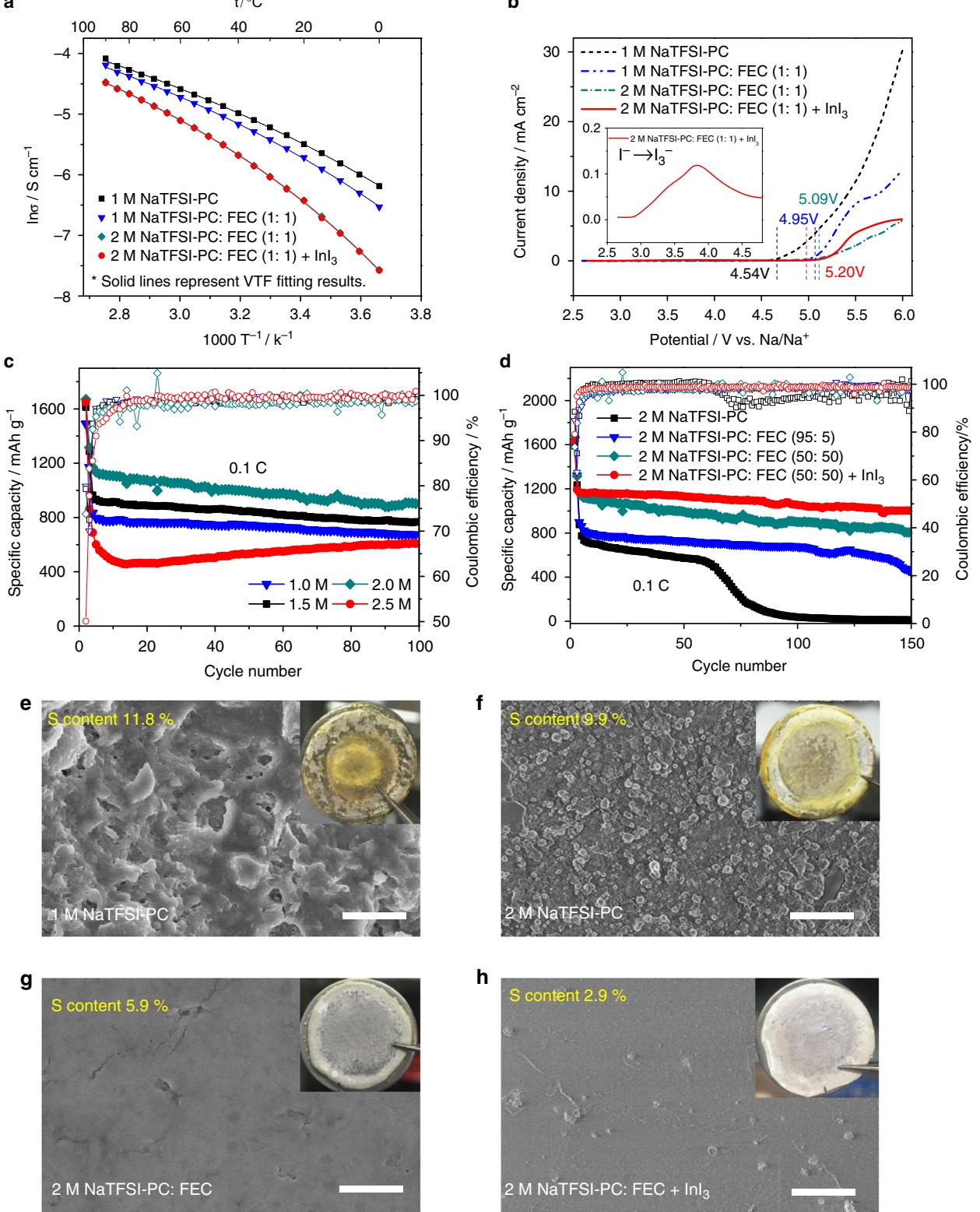

**Fig. 2** Characterization of electrolytes. **a** Ionic conductivities of 1 M NaTFSI in PC, 1 M NaTFSI in PC: FEC (1:1 by volume), 2 M NaTFSI in PC: FEC (1:1 by volume) and 2 M NaTFSI in PC: FEC (1:1 by volume) with 10 mM InI$_3$ samples. The plots represent the experimental data while the solid lines represent VTF fitting results. **b** LSVs of the above four electrolyte samples at a scan rate of 0.1 mV s$^{-1}$ using stainless steel as the working electrode, and Na as the counter and the reference electrodes. **c** Cycling performances of Na/S@MPCF cells using electrolytes with various concentration of NaTFSI in PC: FEC (1:1 by volume) solvents at 0.1 C, and **d** Na/S@MPCF cells containing 2 M NaTFSI in PC: FEC solvents with various FEC proportion and with/without InI$_3$ additive at 0.1 C. The corresponding Coulombic efficiency represented by hollow symbols. **e–h** The FE-SEM images and optical images (shown in inset) of Na anodes obtained from Na/S@MPCF cells using different electrolytes after 50 cycles at 0.1 C. Scale bars are 2 μm in Fig. 2e–h

deliver an ionic conductivity of $1.95 \times 10^{-3}\,\mathrm{S^{-1}\,cm^{-1}}$ at 25 °C, which is sufficient to meet the requirement for room-temperature Na–S batteries. The electrochemical stability of electrolytes was evaluated using linear sweep voltammetry (LSV) on stainless steel electrodes. As shown in Fig. 2b, no peak or noticeable oxidation current is observed in the voltammogram of the 2 M NaTFSI in PC: FEC (1:1 by volume) electrolyte up to 5.09 V vs. Na/Na$^+$. This implies that such electrolyte is stable up to 5.0 V, which is obviously higher than that of 1 M NaTFSI in PC (4.54 V) and 1 M NaTFSI in PC: FEC (1: 1 by volume) (4.95 V). This enhanced electrochemical stability is ascribed to the strong resistance of FEC solvent to oxidation[42], and the pre-formed passive layer attributed to high salt concentration[43]. A small peak starts at around 2.9 V in the voltammogram of 2 M NaTFSI in PC: FEC with InI$_3$ additive (Fig. 2b, inset), which can be assigned to the oxidation of I$^-$ to I$_3^-$[39], and the electrochemical working window slightly increases to 5.20 V. In Supplementary Fig. 8, Na/Na symmetrical cells were subject to galvanostatic cycling measurements at a current density of 0.1 mA cm$^{-2}$ to investigate the compatibility of 2 M NaTFSI in PC: FEC with InI$_3$ electrolyte with Na metal anode. It shows a much smaller overpotential with negligible voltage fluctuation than the cell using 1 M NaTFSI in PC electrolyte (shown in inset) during the 900 h cycles, indicating a uniform Na deposition with a stable electrolyte/Na metal interface[44].

The cycling performances of Na/S@MPCF cells using different concentrations of NaTFSI salt in PC: FEC (1: 1 by volume) as electrolytes are shown in Fig. 2c. The cycling capacities of Na/S@MPCF cells continuously increase with the rising salt concentration (from 680 mAh g$^{-1}$ in 1 M NaTFSI in PC: FEC to 907 mAh g$^{-1}$ in 2 M NaTFSI in PC: FEC after 100 cycles at 0.1 C, more details shown in Supplementary Fig. 9), which can be ascribed to the suppression of Na polysulfide dissolution and inhibition of anodic dendrite formation in concentrated electrolyte[41]. However, the capacity gradually decreases at high salt concentrations beyond 2 M (607 mAh g$^{-1}$ in 2.5 M NaTFSI in PC: FEC after 100 cycles at 0.1 C) owing to excessive electrolyte viscosity (Supplementary Fig. 7). Therefore, an optimal salt concentration is set as 2 M. More remarkably, the cycling capacity of Na/S@MPCF cells consistently increased with increasing FEC proportion in 2 M NaTFSI in PC: FEC electrolytes (from 20 mAh g$^{-1}$ with pure PC to 814 mAh g$^{-1}$ with PC/FEC at a volume ratio of 1:1 after 150 cycles at 0.1 C, Fig. 2d). This could be attributed to the FEC solvent making significant contributions towards restricting the dissolution of Na polysulfides (details shown in Supplementary Fig. 10) and forming a stable protective SEI on the Na anode (see analysis below). However, it should be noted that the increased viscosity (Supplementary Fig. 7) and decreased conductivity (Supplementary Fig. 11) in electrolytes with excessively high FEC ratio (>50%) gives rise to a decline in reversible capacity (Supplementary Fig. 12). Therefore, the FEC proportion was optimized as 50% in this study. It is also shown in Fig. 2d and Supplementary Fig. 13 that the addition of InI$_3$ additive can significantly enhance the Coulombic efficiency and cycling stability of Na–S batteries. The Na/2 M NaTFSI in PC: FEC (1:1 by volume) with 10 mM InI$_3$/S@MPCF cell delivers an initial Coulombic efficiency of 79.1% and a discharge capacity of 1000 mAh g$^{-1}$ after 150 cycles at 0.1 C, which is much higher than the cell without InI$_3$ additive (73.8% and 814 mAh g$^{-1}$). As shown in the inset of Fig. 2b, I$^-$ is oxidized to I$_3^-$ at around 2.9 V vs. Na/Na$^+$ during the charge process. The I$_3^-$ subsequently reacts with Na$_2$S to form Na polysulfides (details shown in Supplementary Fig. 14):

$$n\mathrm{Na_2S} + (n-1)\mathrm{I_3^-} \rightarrow 3(n-1)\mathrm{I^-} + \mathrm{Na_2S}_n \qquad (4)$$

Such reaction can effectively promote the kinetics of Na$_2$S transformation and prevent it from depositing on the cathode.

Furthermore, the In$^{3+}$ can construct a protective In metal layer on the anode in the charge process before Na$^+$ deposition (−2.71 V vs. SHE)[45,46]:

$$\mathrm{In^{3+}} + 3e^- \rightarrow \mathrm{In}\,(-0.340\,\mathrm{V}\;vs.\;\mathrm{SHE}) \qquad (5)$$

Therefore, the anodic corrosion caused by the shuttle effect can be effectively restrained. Hence, the InI$_3$ additive contributes to the improved Coulombic efficiency and outstanding cycling performance.

Figure 2e–h further show field emission scanning electron microscopy (FE-SEM) images of the Na anodes disassembled from the cells used 1 M NaTFSI in PC (cycling performance shown in Supplementary Fig. 15), 1 M NaTFSI in PC: FEC (1:1 by volume), 2 M NaTFSI in PC: FEC (1:1 by volume) and 2 M NaTFSI in PC: FEC (1:1 by volume) with 10 mM InI$_3$ electrolytes after 100 cycles at 0.1 C. It can be clearly seen that massive dendrite structures and holes appear on the surface of Na electrodes obtained from the cells using 1 M NaTFSI in PC electrolyte (Fig. 2e and the cross-sectional FE-SEM image in Supplementary Fig. 16), and the sulfur content on this anode is as high as 11.8 wt% (Supplementary Fig. 17). With the addition of high-concentration salt, FEC solvent and InI$_3$ additive, as expected, the surfaces of Na anodes become smoother and dendrite growth is dramatically inhibited. The sulfur content on the anode of the cell using 2 M NaTFSI in PC: FEC with InI$_3$ is as low as 2.9 wt% (Fig. 2h), demonstrating a significant inhibition of polysulfide shuttling.

X-ray photoelectron spectroscopy (XPS) measurements were performed to investigate the surface components of these Na anodes. As shown in Fig. 3a, the peaks at about 288.5, 286.7, and 284.8 eV in C 1 s can be assigned to O–C=O, C–O, and C–C, respectively. Peaks at 686.6 eV and 683.8 eV in F 1 s are related to C-F in TFSI$^-$ and sodium fluoride (NaF), and 170, 161, and 159.5 eV in S 2p correspond to O=S=O on TFSI$^-$, S$_2^{2-}$, and S$^{2-}$ derived from Na$_2$S$_2$ and Na$_2$S, respectively. Peaks corresponding to SO$_4^{2-}$ at about 168 eV, SO$_3^{2-}$ at about 166.5 eV, S$_8$ at about 163.5 eV (S 2p$_{3/2}$), and 164.7 eV (S 2p$_{1/2}$) are also observed in S 2p spectra[47,48]. There is a general tendency that the C–F bond in F 1 s at about 688 eV[49] and C 1 s at about 292.5 eV[50] become stronger with increasing FEC proportion in electrolytes along with some polycarbonates (poly(CO$_3$)) appearing at 290–291 eV[51] in the C 1 s spectra. The peak intensity of NaF in F 1 s also gradually increases with increasing FEC proportion and salt concentration, which verifies the formation of a F-rich SEI layer on the anode surface. This can be further confirmed by the elemental mapping in Supplementary Fig. 17, Such F-containing components in the SEI are known to have high mechanical strength (e.g., NaF possesses a shear modulus of 31.4 GPa, more than 10 times higher than that of Na metal[12]), which enables the SEI layer to suppress the dendritic growth of Na metal. The S$^{2-}$ and S$_2^{2-}$ peaks in S 2p spectra sharply decline when FEC proportion or salt concentration is increased. This is clear evidence that the solubility of Na polysulfides in FEC or in concentrated electrolyte is so low that only trace amounts of Na$_2$S and Na$_2$S$_2$ are depositing on the Na anode. Moreover, the Na anode of the cell using 2 M NaTFSI in PC: FEC with 10 mM InI$_3$ additive electrolyte exhibits peaks of In 3d at about 457 eV and 445 eV[45] with a In content of 4.8 wt% (Supplementary Fig. 17), meanwhile the peaks of S$^{2-}$ and S$_2^{2-}$ peaks in S 2p spectra nearly disappear, which indicates that the In layer effectively inhibits shuttle effect, as well as I$^-$/ I$_3^-$ scavenges the polysulfides at the cathode. Visual observations on the same amount of sulfur powder together with a Na electrode soaked in different electrolytes were performed to identify the formation and

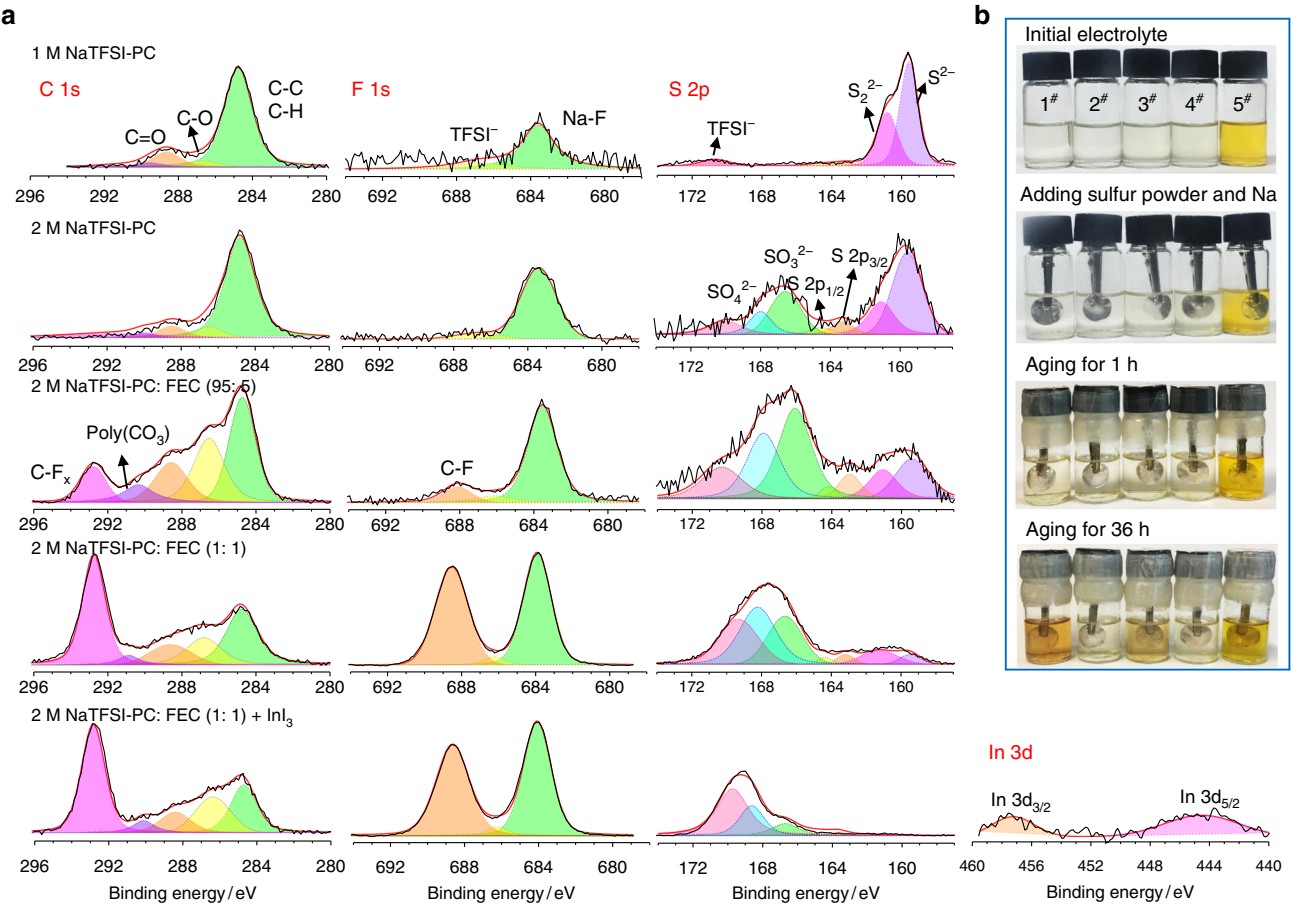

**Fig. 3** Characterization of polysulfides formation in different electrolyte systems. **a** XPS spectra of the Na metals from Na/S@MPCF cells using different electrolytes after 50 cycles at 0.1 C; **b** Visual observation of Na polysulfides formation in five electrolyte samples (1#:1 M NaTFSI in PC; 2#: 1 M NaTFSI in PC: FEC (1:1 by volume); 3#: 2 M NaTFSI in PC; 4#: 2 M NaTFSI in PC: FEC (1:1 by volume); 5#: 2 M NaTFSI in PC: FEC (1:1 by volume) with 10 mM InI$_3$) along with aging time at 60 °C, The same amounts of sulfur (5 mg) and Na metal foils were added into the electrolytes to simulate the self-discharge processes

diffusion of Na polysulfides by observing the color change directly. The electrolyte 1# (1 M NaTFSI in PC) becomes dark in color after aging at 60 °C for 36 h (Fig. 3b). This conspicuous color change demonstrates that the sulfur powder continuously dissolves in the simplistic electrolyte and electrochemically reacts with Na metal to form highly soluble polysulfides with dark colors (corresponding to a self-discharge phenomenon in cells)[52]. However, the 2# (1 M NaTFSI in PC: FEC), 3# (2 M NaTFSI in PC), and 4# (2 M NaTFSI in PC: FEC) electrolytes maintain transparent or light yellow after aging due to the low solubility of Na polysulfides in FEC solvent or high salt concentration solution. The 5# electrolyte, 2 M NaTFSI in PC: FEC (1:1 by volume) with 10 mM InI$_3$, maintains a yellow color from the InI$_3$ additive during the aging test. As further verified by ultraviolet (UV)–visible (Vis) spectra in Supplementary Fig. 18, Na polysulfides were barely formed in this electrolyte. In conclusion, it is the synergistic effect of FEC, highly concentrated salt and InI$_3$ additive that remarkably improves the electrochemical performance of Na–S batteries via an effective suppression of Na polysulfides diffusion, an enhanced Na$_2$S conversion and an efficient construction of protective layer on Na anode.

To investigate the chemistry of Na/2 M NaTFSI in PC: FEC (1:1 by volume) with 10 mM InI$_3$/S@MPCF in a battery, ex-situ Raman spectra were measured to track the changes of sulfur species during the discharge/charge processes. As shown in Fig. 4a, when the initial cathode is discharged to 2.7 V, a sharp

peak appears at about 746 cm$^{-1}$ due to the formation of Na$_2$S$_x$ ($x = 4$–8)[53], which gradually weakens during following discharge process because of its conversion to Na$_2$S and Na$_2$S$_2$. Meanwhile, after discharging to 1.6 V, the Raman spectra show a peak at about 484 cm$^{-1}$ related to Na$_2$S$_4$[54] along with the disappearance of the peaks at 80, 156, 220 and 475 cm$^{-1}$ belonging to S$_8$[55]. The peak of Na$_2$S at 188 cm$^{-1}$ and Na$_2$S$_2$ at 430 cm$^{-1}$ are readily formed at 1.2 V[55], and persistently exist during following charge and discharge process, demonstrating that the solid Na$_2$S formed in the initial discharge process could not be completely oxidized in the following cycles due to poor kinetics during the transitions of solid-state short-chain polysulfides or Na$_2$S[37]. The subsequent charge process as an opposite process shows the peak of Na$_2$S$_x$ ($x = 4$–8) arising from 1.0 V and gradually increasing until 1.8 V. The peaks of S$_8$ appear at 2.2 V in this charge process. It is noteworthy that the peak of Na$_2$S$_x$ ($x = 4$–8) retains its intensity at 2.8 V, indicating the transformation from Na$_2$S to Na$_2$S$_x$ in the presence of I$_3^-$ as mentioned before.

The cyclic voltammetry (CV) of Na/2 M NaTFSI in PC: FEC (1:1 by volume) with 10 mM InI$_3$/S@MPCF cell is shown in Fig. 4b. During the initial cathodic scan, the current slope starts at around 2.1 V vs. Na/Na$^+$, corresponding to the solid–liquid transition from sulfur to dissolved Na$_2$S$_x$ ($x = 4$–8), and the peak at 1.2–0.8 V is related to the formation of Na$_2$S and Na$_2$S$_2$ according to the Raman spectra. The peaks of Na$_2$S and Na$_2$S$_2$ show relatively low repeatability in the following cathodic sweep,

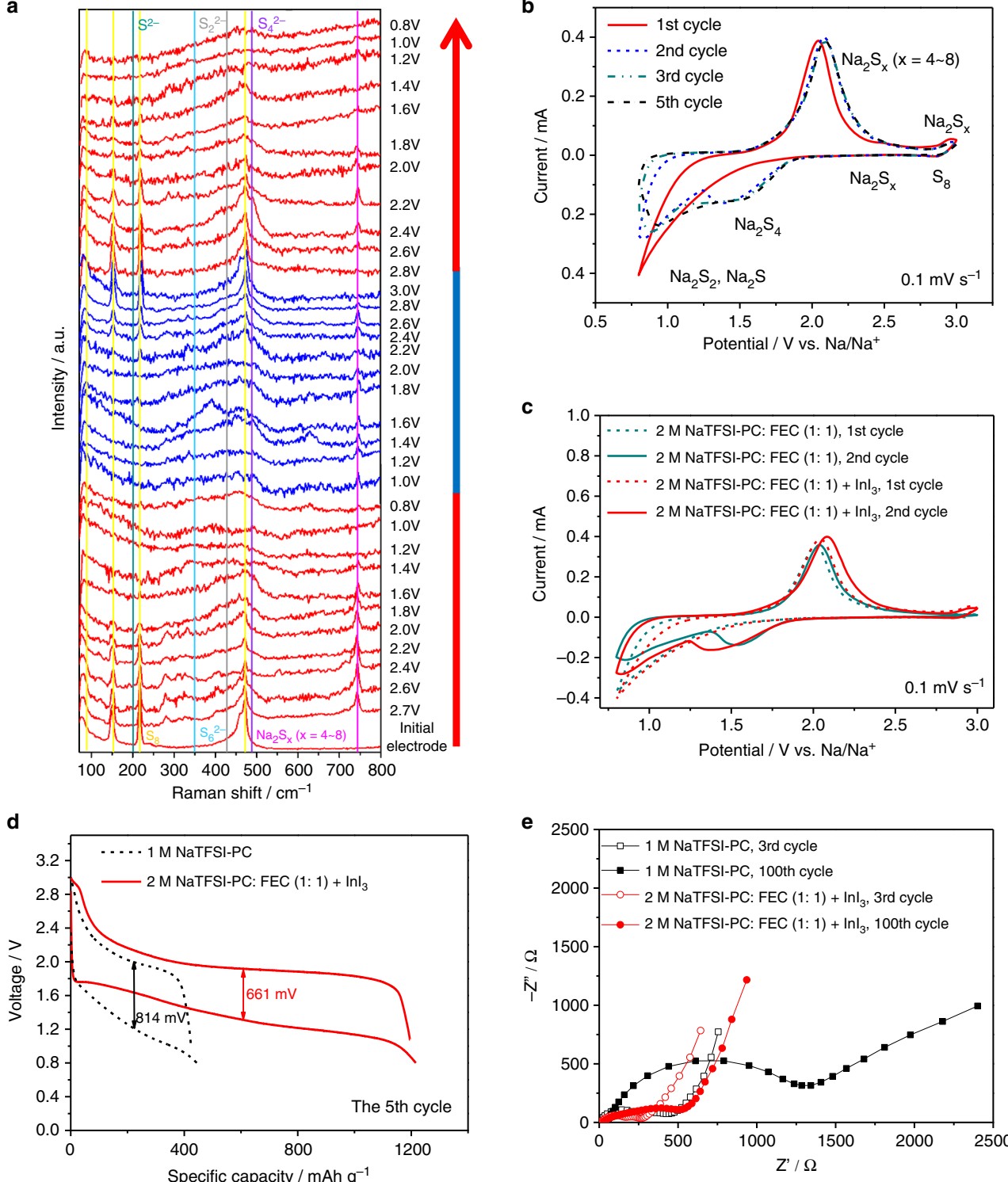

**Fig. 4** Electrochemical behavior of different Na–S battery systems. **a** Ex-situ Raman spectra of the S@MPCF electrodes obtained from Na/2 M NaTFSI in PC: FEC with 10 mM InI$_3$/S@MPCF cells at different charge/discharge potentials; **b** The 1st, 2nd, 3rd, and 5th CV cycles of Na/2 M NaTFSI in PC: FEC with InI$_3$/S@MPCF cell at 0.1 mV s$^{-1}$; **c** CV curves of Na/S@MPCF cells using 2 M NaTFSI in PC: FEC and 2 M NaTFSI in PC: FEC with InI$_3$ electrolytes at 0.1 mV s$^{-1}$; **d** The 5th charge/discharge profiles of Na/S@MPCF cells using 1 M NaTFSI in PC and 2 M NaTFSI in PC: FEC with InI$_3$ electrolytes at 0.1 C; **e** Nyquist plots of Na/S@MPCF cells using 1 M NaTFSI in PC and 2 M NaTFSI in PC: FEC with InI$_3$ electrolytes after three cycles and 100 cycles at 0.1 C, measured in the half charged state

indicating their incomplete conversion. For the anodic scan, a repeatable peak at around 2.1 V presents over all the cycles, corresponding to the transformation of short-chain sodium sulfides to long-chain polysulfides[8]. A small anodic peak is

clearly observed at about 2.9 V in the cell with InI$_3$ additive, which corresponds to the conversion from I$^-$ to I$_3^-$. The addition of InI$_3$ additive greatly reduces the irreversibility of the Na$_2$S and Na$_2$S$_2$ peaks compared to the cell without InI$_3$ additive (Fig. 4c),

which is well consistent with the improved initial Coulombic efficiency in Fig. 2d. The subsequent 2–5th CV curves with two evolving cathodic peaks at around 1.5 V ($Na_2S_x$ ($x = 4$–$8$)) and 0.8 V ($Na_2S$ and $Na_2S_2$) are highly repeatable, indicating that the cathode and anode are highly reversible in the 2 M NaTFSI-PC: FEC (1:1 by volume) with $InI_3$ electrolyte system.

**Electrochemical mechanism discussion**. Supplementary Fig. 19 shows the representative charge/discharge profiles of Na/2 M NaTFSI in PC: FEC (1:1 by volume) with 10 mM $InI_3$/S@MPCF cell at 0.1 C. It is seen that a sloping plateau from 1.8 to 1.5 V and a long plateau in the range of 1.5 to 1.0 V appears in the initial discharge process, which is consistent with the CV curves. The large initial discharge capacity (1635 mAh g$^{-1}$) indicates a high utilization of sulfur in the cathode. It is worth noting that the low-voltage plateau cannot be fully reversed in the following cycles caused by the incomplete $Na_2S$ conversion. However, the initial Coulombic efficiency of 2 M NaTFSI in PC: FEC with $InI_3$ electrolyte (79.1%) is significantly higher than that of 1 M NaTFSI in PC (68.9%) due to the effect of $InI_3$ additive, corresponding to the CV curves of Supplementary Fig. 20. Furthermore, the charge/discharge potential gap of the cell used 2 M NaTFSI in PC: FEC with $InI_3$ electrolyte at 0.1 C (~661 mV) is much smaller than that using 1 M NaTFSI in PC electrolyte (~814 mV) (Fig. 4d). The reduced potential gap suggests that the 2 M NaTFSI in PC: FEC with $InI_3$ electrolyte significantly decreases the polarization of Na–S batteries. Electrochemical impedance spectroscopy (EIS) was measured to evaluate the interfacial behavior and reversibility of cells using such optimized electrolyte. Figure 4e shows the EIS results of the Na–S cells using different electrolytes after different cycles. Such EIS spectra are simulated via an equivalent circuit shown in Supplementary Fig. 21, and the simulation results are summarized in Supplementary Table 3. It is seen that the interfacial resistance ($R_f$) of the cell employed 1 M NaTFSI in PC electrolyte sharply increases after 100 cycles compared with the 3rd cycle (from 156.2 Ω to 789.2 Ω) due to the unstable SEI film caused by the shuttle effect in the cathode and sodium dendrites on the anode as illustrated above. The greatly increased charge transfer resistance ($R_{ct}$, from 372.3 Ω to 856.2 Ω) can be interpreted by the irreversibility of electrically insulating $Na_2S$, which deposited on the surface of electrodes and acts as a barrier for the electron/ion transport. In sharp contrast, the value of $R_{ct}$ and $R_f$ in the cell using 2 M NaTFSI in PC: FEC with $InI_3$ electrolyte are much smaller, and stay almost unchanged during cycling. This indicates an increased conversion degree of $Na_2S$, a suppression of shuttle effect and stability of the electrode/electrolyte interface, and contributes to the dramatically enhanced performance shown in Fig. 2d.

**First-principle calculations of the interactions**. First-principle calculations were employed to further analyze the interaction between Na polysulfide/$Na_2S$ and cathode components as well as electrolyte solvents. As shown in Fig. 5a, the binding energy between $Na_2S_6$ as a representative of Na polysulfides and PC is calculated to be −1.57 eV, which is remarkably stronger than that of $CMC^-$ ion-$Na_2S_6$ (−1.26 eV, Fig. 5g), graphitized carbon matrix-$Na_2S_6$ (−1.46 eV, Fig. 5e), and the formation energy of $Na_4S_{12}$ clusters (−1.29 eV, Fig. 5i). As a result, $Na_2S_6$ tends to dissolve in the PC-based electrolyte. In sharp contrast, the binding energy between $Na_2S_6$ and FEC is as low as −1.22 eV (Fig. 5c), which is obviously lower than that of $CMC^-$ ion-$Na_2S_6$ and graphitized carbon matrix-$Na_2S_6$. Therefore, in this case, $Na_2S_6$ molecules preferentially adhere to the cathode surface rather than dissolves into the FEC-based electrolyte, which is well coincident with the experimental results (Fig. 3b). It is also seen

that the formation energy of $Na_4S_2$ cluster (−2.11 eV, Fig. 5j) is much higher than the binding energies of $Na_2S$ with the solvents (−1.36 eV for PC-$Na_2S$ (Fig. 5b) and −1.09 eV for FEC-$Na_2S$ (Fig. 5d)) and cathode components (−1.16 eV for $CMC^-$-$Na_2S$ (Fig. 5h) and −1.19 eV for graphitized carbon-$Na_2S$ (Fig. 5f)). This clearly confirms that $Na_2S$ tends to agglomerate into a solid network in the electrode, which leads to the difficulty to be transferred to Na polysulfides as illustrated in Fig. 4a. The above theoretical results support an in-depth understanding of Na polysulfide shuttle mechanism in carbonate-based electrolytes.

**Electrochemical performance evaluation**. Figure 6a shows the long-term cycling performance of the Na/2 M NaTFSI in PC: FEC (1:1 by volume) with 10 mM $InI_3$/S@MPCF cells at 0.5 and 1 C, respectively (the cycling performance at 0.1 C is exhibited in Supplementary Fig. 22). The initial irreversible conversion of $Na_2S$ results in an initial Coulombic efficiency of 79.1% at 0.1 C, 71.9% at 0.5 C and 61.6% at 1 C. During the following cycles, high Coulombic efficiency (98.5%–100.4%) and limited capacity decay were achieved. After 200 cycles at 0.1 C, the discharge capacity of Na/2 M NaTFSI in PC: FEC with $InI_3$/S@MPCF cell is 927 mAh g$^{-1}$ with a capacity retention of 77.7% except for the initial cycle (Supplementary Fig. 22). Considering a mid-value discharge voltage of ~1.4 V, the corresponding energy density is around 1477 Wh kg$^{-1}$ and 886 Wh kg$^{-1}$ calculated based on the mass of sulfur and S@C composite, respectively. Furthermore, after about 500 cycles at 0.5 and 1 C, discharge capacities of 648 mAh g$^{-1}$ and 581 mAh g$^{-1}$ remained, respectively, which demonstrates an outstanding long cycling stability. The rate performances of the Na/2 M NaTFSI in PC: FEC with $InI_3$/S@MPCF cells are shown in Fig. 6b, while the corresponding discharge/charge curves are presented in Supplementary Fig. 23. The Na/2 M NaTFSI in PC: FEC with $InI_3$/S@MPCF cell delivers specific charge capacities of 1170 mAh g$^{-1}$, 1107 mAh g$^{-1}$, 984 mAh g$^{-1}$, 867 mAh g$^{-1}$, and 699 mAh g$^{-1}$ at 0.1, 0.2, 0.5, 1, and 2 C, respectively, which are much higher than the cell using 1 M NaTFSI in PC electrolyte. Furthermore, the capacity of the Na/2 M NaTFSI in PC: FEC with $InI_3$/S@MPCF cell successfully recovers to 1140 mAh g$^{-1}$ (97.4% of that in the 5th cycle) when the current density is switched back to 0.1 C, reflecting that this novel Na–S battery system is robust and highly stable. The cycling performances of the Na/2 M NaTFSI in PC: FEC with 10 mM $InI_3$/S@MPCF cells with high sulfur loadings were further investigated. As shown in Fig. 6c, capacities of 1134 mAh g$^{-1}$, 1038 mAh g$^{-1}$, 1007 mAh g$^{-1}$, 354 mAh g$^{-1}$, and 301 mAh g$^{-1}$ are successfully retained after 50 cycles at 0.1 C with sulfur loadings of 0.35, 1.24, 1.57, 4.27 and 4.64 mg cm$^{-2}$, respectively, which meets the requirement of practical applications. This new electrolyte also shows excellent cycling performance in Na–S cells using other S@porous carbon composite electrodes (such as S@CMK-3 shown in Supplementary Fig. 24a).

As shown in Fig. 6d, the electrochemical performance of the Na/2 M NaTFSI in PC: FEC (1:1 by volume) with 10 mM $InI_3$/ S@MPCF cells in this work is superior to most previously reported Na–S cells. As an overview of the state-of-the-art, Fig. 6e shows a performance comparison of the as-developed Na–S batteries with other previously reported Li and Na battery systems. In most cases, the practical specific capacity and energy density (based on the mass of cathodic active material only) for Na batteries is less than 900 mAh g$^{-1}$ and 1100 Wh kg$^{-1}$ (specially, <500 mAh g$^{-1}$ and <900 Wh kg$^{-1}$ for high-temperature Na–S batteries[7]). The practical specific capacity and energy density of the room-temperature Na–S battery in this work not only surpass these Na battery systems, but also exceed the traditional lithium-ion battery systems using cathode

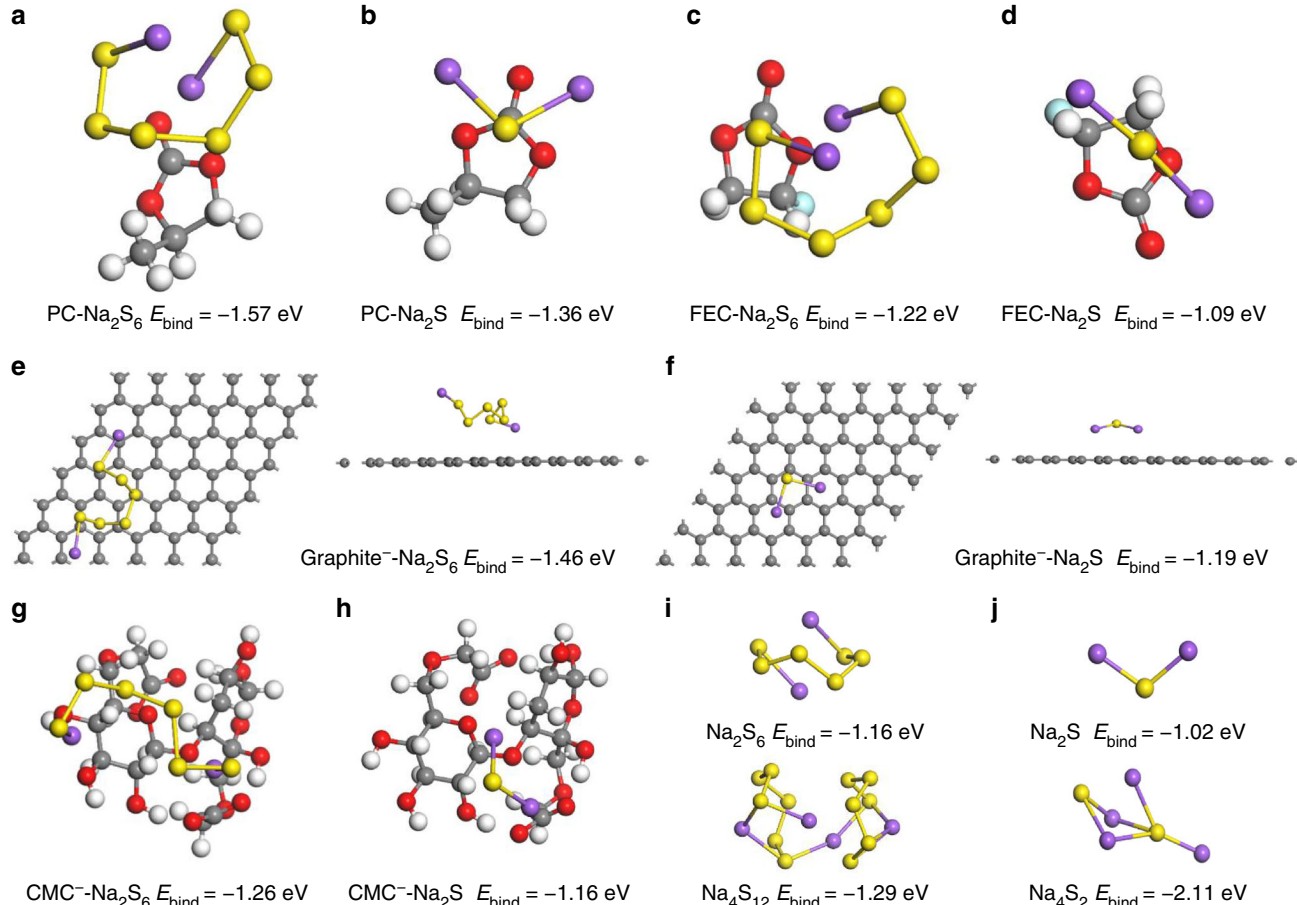

**Fig. 5** First-principle simulations. **a–d** Calculated binding energies of **a** $Na_2S_6$ and **b** $Na_2S$ with PC solvent, and **c** $Na_2S_6$ and **d** $Na_2S$ with FEC solvent; **e**, **f** Binding energies of **e** $Na_2S_6$ and **f** $Na_2S$ on graphitized carbon from top view (left) and side view (right); **g, h** Binding energies of **g** $Na_2S_6$ and **h** $Na_2S$ with $CMC^-$ ion; **i, j** Binding energies of the most stable **i** $Na_2S_6$ molecular/$Na_4S_{12}$ cluster, and **j** $Na_2S$ molecular/$Na_4S_2$ cluster. Yellow, purple, gray, white, red and blue balls represent sulfur, sodium, carbon, hydrogen, oxygen and fluorine atoms, respectively

materials such as lithium iron phosphate (LFP), lithium manganese oxide (LMO), and lithium cobalt oxide (LCO).

## Discussion

In summary, we discovered that a multifunctional electrolyte containing 2 M NaTFSI in PC: FEC (1:1 by volume) co-solvents with $InI_3$ additive could greatly enhance the reversible capacity and cyclability of room-temperature Na–S batteries. The high salt concentration in this electrolyte effectively reduces the solubility of the Na polysulfides and simultaneously stabilizes the Na anode. The FEC solvent not only possesses a low binding energy with Na polysulfides, which successfully enables polysulfides to remain in the cathode instead of dissolving into electrolyte, but also benefits the formation of a stable F-rich SEI film on the Na metal surface upon cycling. The $InI_3$ additive acts as redox mediator to greatly improve the Coulombic efficiency of batteries by enhancing the transformation kinetics of $Na_2S$ in the cathode, and also forms a protective In layer on the Na anode against polysulfide corrosion. Such "cocktail optimized" electrolyte allows stable cycling of room-temperature Na–S batteries with high-energy density. These key findings open up a new direction to inspire revolutionary improvements in the performance of room-temperature Na–S batteries. This electrolyte design strategy can also be extended to a wide range of Na-based rechargeable battery systems (e.g., Na–oxygen, Na–selenium, and Na–iodine batteries), and boost the development of low-cost and high-performance energy storage devices.

## Methods

**Preparation and characterization of electrolytes**. Bis(trifluoromethane)sulfonimide sodium salt (NaTFSI, DoDoChem, 99.8%), lithium bis(trifluoromethanesulfonyl)imide (LiTFSI, DoDoChem, 99.8%) and indium triiodide ($InI_3$, Sigma Aldrich, 99.998%) additives were fully dried at 80 °C for 24 h before use. The employed electrolyte solvents contained triethylene glycoldimethyl ether (TEGDME, DoDoChem, 98%), propylene carbonate (PC, DoDoChem, 99.98%) and fluoroethylene carbonate (FEC, DoDoChem, 99.95%). All procedures for electrolyte preparation were carried out in an Ar-filled glove box (MBraun) with the concentrations of moisture and oxygen below 0.5 ppm.

The ionic conductivities of the electrolytes were measured by electrochemical impedance spectrum (EIS) from 100 kHz to 1 Hz with an alternating current amplitude of 5 mV on a VMP3 multichannel electrochemical station (Bio Logic Science Instruments, France). The test cells were assembled by soaking two stainless steel blocking electrodes in electrolyte samples. Prior to the conductivity measurements, the cells were kept at each test temperature (from 0 to 90 °C) for 30 min to reach thermal equilibrium. The electrochemical stability windows of the electrolytes were determined by linear sweep voltammograms performed on Na/stainless steel cells at 25 °C. The LSVs were measured from open circuit potential to 6 V (vs. $Na^+/Na$) at a scan rate of 5 mV $s^{-1}$ on the VMP3 electrochemical station. To evaluate the compatibility of electrolyte with Li or Na metal, galvanostatic cycling measurements consisting of repeated 2-h charge and 2-h discharge cycles were performed on a symmetrical Li/Li or Na/Na cells at 0.1 mA $cm^{-2}$. The Na polysulphide dissolution experiments were carried out as follows: $Na_2S$ and S with a mole ratio of 1:7 (1.3 and 3.7 mg) were mixed and added in 10 mL electrolyte solvents, and kept at 60 °C to record the color changes along with the aging time. Corresponding UV–Vis spectra were collected with a SEC 2000 UV–Visible spectrophotometer (ALS Co., Ltd.). The viscosity of the electrolytes was performed by the Ubbelohde viscometer (Minbo Co., Ltd.) at 25 °C.

**Assembly and characterization of Na-S batteries**. The preparation for multiporous carbon fibers (MPCFs) is provided in Supplementary Information. The cathode material, S@MPCF, was prepared following a melt-diffusion strategy.

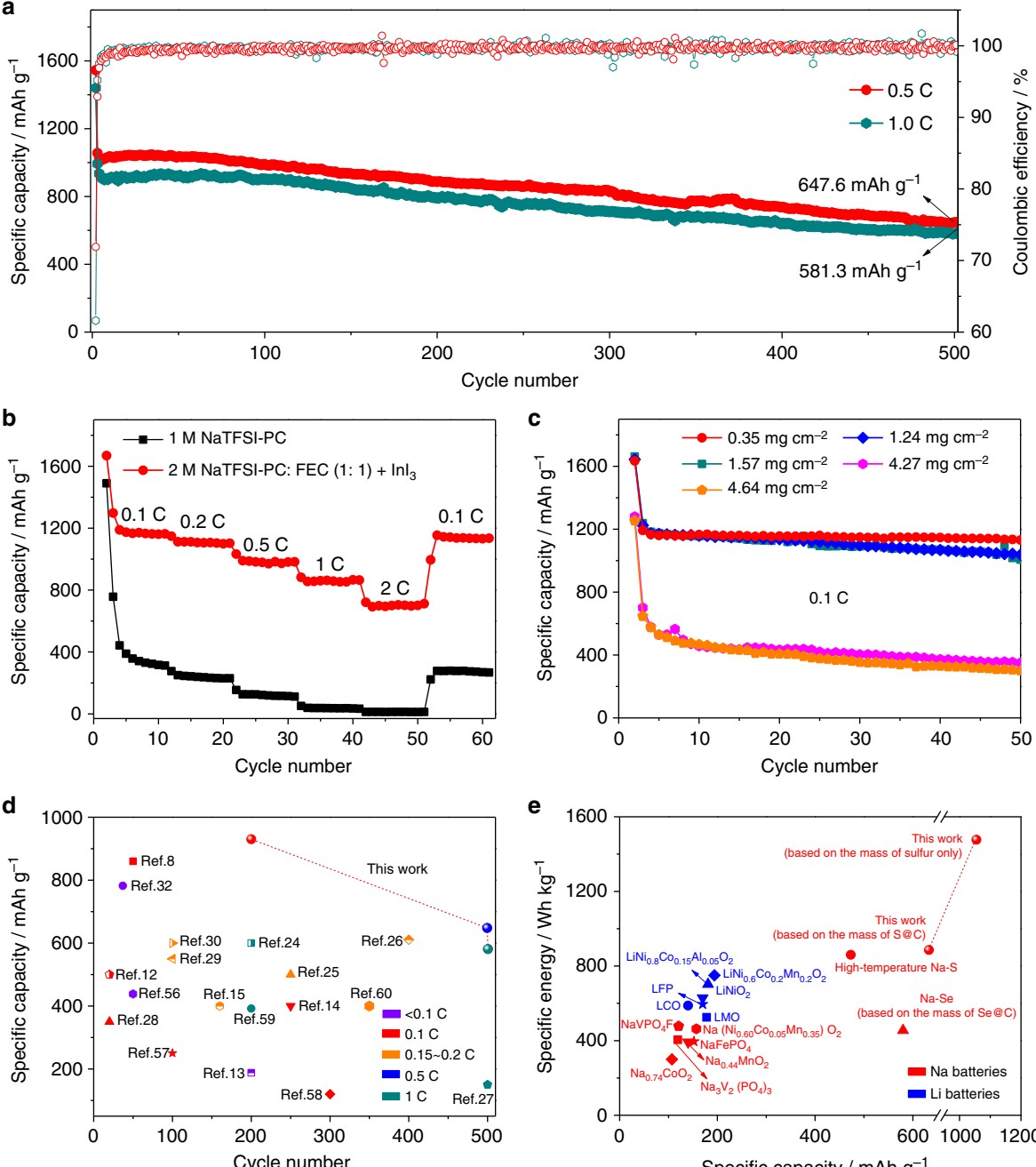

**Fig. 6** Electrochemical performances of Na/2 M NaTFSI in PC: FEC with 10 mM InI$_3$/S@MPCF cells. **a** Long-term cycling performances of Na/2 M NaTFSI in PC: FEC (1: 1 by volume) with 10 mM InI$_3$/S@MPCF cells at 0.5 C and 1 C; **b** Rate performances of Na/S@MPCF batteries using 1 M NaTFSI in PC and 2 M NaTFSI in PC: FEC with InI$_3$ electrolytes. The sulfur loading in Fig. 6a, b is ~0.36 mg cm$^{-2}$; **c** Cycling performances of Na/2 M NaTFSI in PC: FEC with InI$_3$/S@MPCF cells at 0.1 C with different sulfur mass loadings; **d** Comparison of practical specific capacities and cycling performances for representative reported room-temperature Na-S batteries and this work[8,12–15,24–30,32,37,38,57–61]; **e** Comparison of practical specific capacities and energy densities of Na batteries and Li batteries with representative reported cathode materials[7,62–64]. The specific capacities and energy densities are evaluated by the mass of cathodic active materials only

MPCFs and nano sulfur powder (Dk Nano technology, Beijing) were ground together at a weight ratio of 4:6, and subsequently the S/MPCF mixture in a sealed container were heated at 155 °C for 10 h and further heated at 300 °C for 1 h in Ar. The S@MPCF electrodes were prepared by a slurry-coating method. The slurry composed of 80 wt% S@MPCF, 10 wt% Super-P as conductive agent, and 10 wt% sodium carboxymethyl cellulose (CMCNa, Macklin, 1500–3100 mpa.s) as binder in deionized water was coated onto a carbon-coated aluminum foil and then dried at 60 °C under vacuum for 12 h. CR2032 coin cells were assembled in an Ar-filled glove box using glass fiber membranes (Whatman GF/A) for Na–S batteries. The sulfur/electrolyte ratio in each cell was uniformly set at ~50 g L$^{-1}$. The assembled

Na–S cells were cycled between 0.8 and 3.0 V at various charge/discharge rates (1 C = 1675 mA g$^{-1}$) on a Land 2001 A battery testing system at 25 °C. The specific capacity values were calculated based on the mass of sulfur, and the Coulombic efficiency calculated as percentage of the charge capacity in respect to the discharge capacity. Cyclic voltammograms (CVs) of the assembled cells were tested using the VMP3 electrochemical working station at a scanning rate of 0.1 mV s$^{-1}$. Electrochemical impedance spectra (EIS) of cells was examined using the VMP3 multi-channel electrochemical station in the frequency range of 10$^{-2}$ to 10$^5$ Hz by applying a disturbance amplitude of 5 mV. The cells after designated cycling tests were transferred into a glove box and dissembled for postmortem analysis. The air-

sensitive electrode samples were rapidly transferred into the vacuum chambers of SEM/XPS/Raman under the protection of vacuum box before the following tests. The ex-situ Raman spectra were obtained with a Lab RAM HR800 (Horiba) using 532 nm incident radiation.

**Theoretical calculations**. All structure relaxation and electronic structure calculations were performed with density functional theory with the projector-augmented wave method. The exchange-correlation functional of Perdew, Burke, and Enzerhof (PBE) was employed to analyze the exchange and correlation potentials. The cutoff energy level was set as 500 eV, the SCF tolerance level was as $1.0 \times 10^{-5}$ au in geometry optimization, while the SCF tolerance was set as $1.0 \times 10^{-6}$ for energy calculation. The semi-empirical London dispersion corrections of Grimme et al.[56] were applied to take the dispersion interactions of van der Waals into consideration. For graphite, a $4 \times 4 \times 1$ super cell with a 20 Å vacuum was used.

## Data availability

The data that support the findings of this study are available from the corresponding author upon reasonable request.

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

## Acknowledgements
We would like to acknowledge the support by National Key Basic Research Program of China (No. 2014CB932400), National Natural Science Foundation of China (No. U1401243), Shenzhen Technical Plan Project (No. JCYJ20150529164918735 and JCYJ20170412170911187), Guangdong Technical Plan Project (No. 2015TX01N011), the Australian Renewable Energy Agency project (ARENA 2014/RND106), and the ARC Discovery Project (DP170100436). We thank Dr. Peng Li from Nanjing University of Aeronautics and Astronautics for conducting the first-principle calculations and Dr. Ming Liu from Delft University of Technology for giving valuable advice on the characterization of polysulfide shuttle phenomenon. D. Shanmukaraj would like to thank IKERMUGIKORTASUNA, Basque mobility grant for a short stay at University of Technology Sydney (UTS), Sydney, Australia.

## Author contributions
B.L., M.A., and G.W. conceived and designed this work; X.X. and D.Z. performed the experiments and wrote the manuscript. X.Q. designed the synthesis of multiporous carbon fibers materials. K.L. contributed to the analysis of Raman spectrum. F.K., D.S., and T.R. discussed the results and participated in the preparation of the paper.

## Additional information

**Competing interests:** The authors declare no competing interests.

