## [Peer Review File · Nature Communications]

Editorial note: The figure found on page 37 ('Figure R2') is reproduced with permission from John Wiley & Sons: Sen Xin, Ya - Xia Yin, Yu - Guo Guo, et al A High - Energy Room - Temperature Sodium - Sulfur Battery (2014): Adv. Materials. <https://doi.org/10.1002/adma.201304126>

Reviewers' comments:

Reviewer #1 (Remarks to the Author):

This manuscript describes the use of a very specific blend of 2M NaTFSI, PC, FEC and InI₃ for enhancing performance of a sodium sulfur battery with the sulfur imbedded microporous carbon host in the cathode. However, as the author mentioned these improvements are "cocktail optimized", or simple combination of previous ideas or techniques reported before in sodium batteries or Li-S systems. For instance, the high concentrated salt for sodium battery was reported by Ji-Guang Zhang's group (Nano Energy 30 (2016) 825–830); the InI₃ additive was applied in Li-S battery for the same functions (Journal of Power Sources 361 (2017) 203-210); the idea of PC-FEC co-solvent was reported in Nano Research 7.10 (2014): 1466-1476. Therefore, this work is an incremental research combining the existing methods rather than a real scientific breakthrough.

In addition, there are a few detailed questions with this manuscript:

1. Please comment on (in the main text) whether the intermediate polysulfide species will react with the carbonate based electrolyte solvent because carbonate solvents are known to be quite susceptible to reaction with polysulfide anions. (J. Phys. Chem. C 2011, 115, 25132–25137). However this was not discussed anywhere in the manuscript.
2. It was demonstrated that polysulfide becomes colorless after injecting into a carbonate-based electrolyte. However, polysulfide species will react with carbonates. Therefore, believe there experiments shown in Figure 3b will be convoluted by the degree of solubility of the reaction product between carbonates and polysulfide. Overall, please comment on the effect of the polysulfide reaction with carbonate (J. Phys. Chem. C 2011, 115, 25132–25137) on Figure 3b.
3. There appears to be a drastic drop in performance when the sulfur areal loading was increased from 1.24 mg cm⁻² to 1.57 mg cm⁻². Please provide an explanation to this phenomenon.
4. What is the areal sulfur loading for the cells tested in Figure 6 A&B. Please label the areal S loading for both Fig 6 A&B.
5. Please show results for sample with just InI₃ added into 1M NaTFSI-PC electrolyte. This control experiment was not shown in the manuscript.
6. The experimental EIS spectrum is not present (or cannot be seen) in Figure S17, please change so it is more apparent.
7. It was mentioned numerous times in the manuscript that an In layer was formed on Na. However, there is not enough evidence to support this claim. Please provide EDS and XPS analysis on the indium and fluorine content on the surface of the sodium metal cycling as shown in Figure 2h. Also, please contrast the morphological difference between the Na metal tested with and without In via a cross-sectional SEM of the Na metal after cycling to support claim of a In layer formation of In.

Based on the above comments, the review believes that the manuscript reports good electrochemical performance and fundamental science study, but it's not suitable for publication in Nature Communication due to the insufficient novelty. I recommend submission to a more specialized journal instead.

Reviewer #2 (Remarks to the Author):

The manuscript described a room-temperature sodium-sulfur battery containing PC/FEC electrolyte with NaTFSi and InI₃. There are a lot of interesting results. It dealt with whole parameters of Na/S cell using sulfur in porous carbon.

1. Sulfur electrode : a. bare sulfur electrode, b. sulfur electrode in multiporous carbon fibers (S@MPCF)

2. Binder : a. CMC, b. PVdF

3. Electrolyte : a. Ether based electrolyte of TEGDME, b. Carbonate based electrolyte : PC, PC-5%FEC, PC-50%FEC – with different concentration of salts NaTFSi and InI3

Also, there are a lot of interesting results such as optimum electrolyte for long term cycle with high capacity, dissolution behavior of sodium polysulfides in FEC electrolyte, and also optimum binder of CMC instead of PVdF.

However, it is very difficult to understand the main concept and difference from previous works and it is necessary to explain clearly data and discussion.

Another issue is close correlation between data in manuscript and previous data.

1. The sulfur electrode of S@MPCF has nanosized pores in Fig. S2(e) and 61.09% sulfur in carbon from TGA of Fig. S2(f). The electrochemical properties critically depend on the position of sulfur whether it is inside nanopore or on the surface. The sulfur inside nanopore can reversibly cycled in carbonate electrolyte but sulfur on the surface shows similar behavior with elemental sulfur, which can only cycle in ether electrolyte such as TEGDME. However, there was no critical explanation about this in the manuscript. In Fig. S4(a) shows irreversible behavior in Li/S cell which is well matched with sulfur on the surface, similar to bare sulfur. Also, this is well matched with Fig. S4(c) in Li/S cell, but it is different from Na/S cell in Fig S4(c) and also other data in Fig.2, which can cycle in carbonate based electrolyte. Since this is very confusing, it should be explained clearly.

2. There is important improvement by optimizing electrolyte in Fig. 2 such as high concentration of salts and FEC and InI3 addition. The electrochemical behavior of basic(or original) sulfur electrode should be confirmed or matched with previous results because there was already good results using sulfur in porous carbon without modification of electrolyte.

Reviewer #3 (Remarks to the Author):

The manuscript entitled "A room-temperature sodium-sulfur battery with high capacity and stable cycling performance" reported an carbonate electrolyte with bifunctional additive (InI3) for sodium-sulfur batteries, in which anion is a redox mediator to improve kinetic of sulfur cathode and cation is able to form robust SEI on anode side. Electrochemical performance of room-temperature Na-S battery improve a lot and the mechanism is discussed with satisfactory characterizations. I would like to recommend its acceptance after my below concern could be addressed.

1) I suggest authors should explain in detail about the cathode. As my understanding, the S@MPCF cathode is a sulfur-PAN composite which can work well in carbonate-based electrolyte because S is completely covalently immobilized in PAN host. The voltage profile shown in Fig. 4d with one plateau around ~1.5 V indicates a solid-solid (S-Na₂S) conversion which is different from the reaction mechanism of that in ether-based electrolyte (solid-intermediate-solid, S-Na₂S_x-Na₂S). That is why this covalent-bonded S composite can work in carbonate-based electrolyte. Generally, based on Li-S chemistry, irreversible reaction can occur between long-chain polysulfides and carbonate electrolytes (ACS Cent. Sci. 2015, 1, 449–455) if S is not covalent-bonded/encapsulated by hosts. Therefore, if the redox reaction in this paper is deemed "solid-solid" reaction, why polysulfides dissolving in the electrolyte/shuttling can occur? Could Authors please clarify the difference between Na-S and Li-S chemistry. This is very important to reveal the mechanism why this electrolyte can work well.

2) As previous work, microporous porous only allow small sulfur molecular parasitic which also guarantee that sulfur is able to work in non-polysulfide dissolute carbonate electrolyte. I am curious about sulfur structure in MPCF with microporous (<1nm) and surprise the high sulfur loading in MPCF

(60 wt%) which is higher than most of microporous carbon (<40 wt% sulfur loading). Could the author explain its unique?

3) To further clarify the influence of concentration, I suggest authors provide the voltage profiles and cycling performance using 1M NaTFSI-PC and 2M NaTFSI-PC for comparison.

4) I notice that the electrolyte/sulfur ratio is 20 $\mu\text{L}/\text{mg}$. I am wondering if the parameter is optimized or not? What is the common value of this ratio in most references for Na-S batteries?

5) Gravimetric energy density of Na-S batteries in Figure 6 e is overestimated that would mislead the reader. The overall capacity of Sulfur-Carbon composition should be used instead of pure sulfur. The 1477 Wh/kg is only based on the sulfur weight, but when turning to gravimetric energy density, it should be $1477 * 60\%$ (sulfur content) = 886 Wh/kg, which is comparable with the current lithium-ion cathodes.

6) I suggest the author supply the first charge-discharge profile which is very important for the reader to get the full picture of Na-Sulfur batteries. And I disagree the author's explanation the huge irreversible capacity loss in the first cycle and low coulombic efficiency " The initial irreversible conversion of Na_2S together with the formation of SEI on the anode result in an initial Coulombic efficiency of 79.1 % at 0.1 C, 71.9 % at 0.5 C and 61.6 % at 1 C (the Coulombic efficiency is calculated as percentage of the charge capacity in respect to the discharge capacity).-----" In the work, the anode is excessive, so SEI formation can not influence efficiency in half cell and the efficiency only express the information about cathode, so I think the irreversible capacity loss should result from cathode. Please give the reasonable explanation?

Response to Reviewers' Comments

Reviewer #1:

This manuscript describes the use of a very specific blend of 2M NaTFSI, PC, FEC and InI₃ for enhancing performance of a sodium sulfur battery with the sulfur imbedded microporous carbon host in the cathode. However, as the author mentioned these improvements are “cocktail optimized”, or simple combination of previous ideas or techniques reported before in sodium batteries or Li-S systems. For instance, the high concentrated salt for sodium battery was reported by Ji-Guang Zhang’s group (Nano Energy 30 (2016) 825–830); the InI₃ additive was applied in Li-S battery for the same functions (Journal of Power Sources 361 (2017) 203-210); the idea of PC-FEC co-solvent was reported in Nano Research 7.10 (2014): 1466-1476. Therefore, this work is an incremental research combining the existing methods rather than a real scientific breakthrough.

Reply: Thank you for your comments, which will further improve the quality of our work. In this work, we provide fundamental scientific study on the electrochemical properties of carbonate-based electrolyte in room-temperature Na-S batteries, and significantly improve the performance of this battery system via electrolyte optimization. Some of the optimization methods in electrolyte have been reported in Na-ion batteries or Li-S batteries. However, these methods have not yet been applied in Na-S batteries. The fundamental mechanisms are intrinsically different, which have not been investigated previously. A point-to-point explanation is listed in detail as below:

1) In the work reported in *Nano Res.* 7, 1466-1476 (2014), FEC was employed as a solvent in the Na-ion batteries with an alloy-based (SnSb) anode. They found that the addition of small amount (5 wt%) of FEC can form a protective film on the Na surface to suppresses the unfavorable side reactions, while a large proportion (50 % by volume) of FEC additive is not desirable. **In our work, we used FEC as a co-solvent for room-temperature Na-S batteries based on a conversion chemistry. The FEC solvent not only benefits the formation of a stable F-rich SEI film on the Na metal surface upon cycling, but also possesses a low binding energy with Na polysulfides which successfully enables polysulfides to remain in the cathode instead of dissolving into electrolyte. A large proportion (50 % by volume) of FEC additive attributes to the best electrochemical performance in Na-S cells.** So, the FEC solvent functions with a totally different electrochemical mechanism in Na-S cells compared with traditional Na-ion batteries. Therefore, there is no similarity between our work and previous reports.

2) In the work reported in *Nano Energy* 30, 825–830 (2016), the electrolyte composed of highly concentrated Na salt in ether solvent enabled a good cycling stability of Na-ion batteries with Na₃V₂(PO₄)₃ cathode via minimizing the side degradation reactions. In our work, we successfully proved that the high salt concentration in the carbonate-based electrolyte not only formed a stable F-rich SEI and a dendrite-free Na surface during cycling, but also effectively reduced the

solubility of the Na polysulfides. This clearly shows that the high concentrated salt in electrolytes follows an obviously different mechanism in Na-S cells compared with Na-ion batteries.

3) In the work reported in *J. Power Sources* **361**, 203-210 (2017), InI_3 was proposed for Li-S batteries. They found that the deposited In layer protected the Li anode from side reactions, and I^-/I^{3-} redox mediator was capable of decomposing side products deposited on the Li anode and separator. In our work, InI_3 additive was applied to overcome the intrinsic issue of Na-S cells, which is the poor transformation kinetics related to the conversion from Na_2S_2 or Na_2S to long-chain polysulfides due to dramatic volume change caused by the large ion size of Na^+ . For the first time, we reveal that the low Coulombic efficiency and rapid capacity fading of S-based cathode in Na-S batteries caused by such poor kinetics can be greatly overcome by redox mediators such as I^-/I^{3-} . The InI_3 can also form a protective In layer on the Na anode against polysulfide corrosion. Therefore, the InI_3 plays different roles in Na-S batteries compared with Li-S batteries.

4) Based on above discussion, it can be clearly seen that in this work, the optimization of electrolyte for Na-S batteries are innovative and essentially different from previously reported techniques. Considering that the research on room-temperature Na-S batteries system is currently in its infancy and only few reports about cyclable Na-S batteries have been published, we believe that this research is a major scientific breakthrough and of great significance to the development of room-temperature Na-S batteries. To highlight the novelty of our work, we have cited these three references and revised the section of Introduction on Page 5 and 9 in the revised manuscript.

In addition, there are a few detailed questions with this manuscript:

1. Please comment on (in the main text) whether the intermediate polysulfide species will react with the carbonate based electrolyte solvent because carbonate solvents are known to be quite susceptible to reaction with polysulfide anions. (J. Phys. Chem. C 2011, 115, 25132–25137). However this was not discussed anywhere in the manuscript.

Reply: Thank you for this precious comment. As indicated by the reviewer, it is well known that in Li-S batteries, the nucleophilic sulfide anions actively react with carbonate solvents via nucleophilic addition or substitution reaction, which results in a rapid capacity fading^[1]. As shown in **Figure S6a**, same amounts of sulfur (10 mg) together with Li or Na foils were added into different electrolytes to observe the formation of polysulfides. After aging for 48 h, the tetraethylene glycol dimethyl ether (TEGDME)-based electrolyte immersing with Li foil obviously turned to dark brown color due to the formation of Li polysulfides; Meanwhile, the propylene carbonate (PC)-based electrolyte immersing with Li foil almost did not change its color, which indicates that Li polysulfides cannot be massively formed in carbonate-based electrolytes due to side reactions. This can be further verified by the images of separators (**Figure S6b**) together with the Raman spectra of sulfur electrodes (**Figure S6c**) obtained from discharged Li-S cells, and also the UV-Vis spectra results (**Figure S6d**). In sharp contrast, it is seen from **Figure S6a** and **Figure S6b** that dark-colored Na polysulfides can be generated in PC-based electrolyte, meanwhile the Raman spectrum of the discharged sulfur electrode from Na/1 M NaTFSI in PC/bare S cell shows clear peaks of Na polysulfides and Na_2S (**Figure S6c**)^[2]. This may be due to

the fact that the larger ionic radius of Na^+ than Li^+ leads to less dissociation in polar solvents, which results in a lower reactivity of Na^+ -polysulfide $^-$ ion pairs than that of Li^+ -polysulfide $^-$ ion pairs^[3,4]. Hence, the side reactions between Na polysulfides and carbonate solvents are much less than those between Li polysulfides and carbonate solvents. When composited with mesoporous carbon to relieve the volume change ($\sim 260\%$) during the cathodic reaction from sulfur to Na_2S ^[5], the S@multiporous carbon fiber (MPCF) electrode can maintain a relatively stable capacity of 349 mAh g^{-1} after 20 cycles at 0.1 C in Na/1 M NaTFSI in PC/S@MPCF cells (**Figure R1**). In contrast, the Li/1 M LiTFSI in PC/S@MPCF cell can only deliver $\sim 1\text{ mAh g}^{-1}$ after 20 cycles at 0.1 C. Based on the above analysis, it can be concluded that the electrochemical performance of Na-S battery is negligibly affected by the side reaction between Na polysulfides with carbonate solvents. We have added this discussion on Page 5~6 and cited the related references in the revised manuscript.

Figure S6 | **a** Visual observation of polysulfides formation in four electrolyte samples (1#:1 M LiTFSI in TEGDME with Li metal foil; 2#: 1 M LiTFSI in PC with Li metal foil; 3#: 1 M NaTFSI in TEGDME with Na metal foil; 4#: 1 M NaTFSI in PC with Na metal foil) along with aging at 60 °C for 48 h. The same amounts of sulfur (10 mg) were added into 3 g electrolyte samples to simulate the self-discharge processes. **b** The optical images of the segregators obtained from 1#: Li/1 M LiTFSI in TEGDME/bare S; 2#: Li/1 M LiTFSI in PC/bare S; 3#: Na/1 M NaTFSI in TEGDME/bare S; 4#: Na/1 M NaTFSI in PC/bare S cells after the initial discharging at 0.1 C.

Figure S6 | c Raman spectra of the bare sulfur electrodes obtained from Li/1 M LiTFSI in TEGDME /bare S, Li/1 M LiTFSI in PC/bare S, Na/1 M NaTFSI in TEGDME/bare S and Na/1 M NaTFSI in PC/bare S cells after an initial discharging at 0.1 C.

Figure S6 | d UV-Vis spectra of different electrolyte samples with same amounts of sulfur powder and Li or Na metal foils after aging at 60 °C for 48 h, corresponding to **Figure S6a**. S_4^{2-} : ~320 nm; S^{2-} and S_2^{2-} : 220~260 nm^[6].

Figure R1 | Cycling performances of Li/1 M LiTFSI in PC/S@MPCF and Na/1 M NaTFSI in PC/S@MPCF cells at 0.1 C.

References:

- [1] Gao, J. *et al. J. Phys. Chem. C* **115**, 25132-25137 (2011).
- [2] Yeon, J. T. *et al. J. Electrochem. Soc.* **159**, A1308-A1314 (2012).
- [3] Bhattacharyya D. N. *et al. J. Phys. Chem.* **9**, 612-623 (1965).
- [4] Bhattacharyya, D. N. *et al. J. Phys. Chem.* **69**, 608-611 (1965).
- [5] Carter, R. *et al. Nano Lett.* **17**, 1863-1869 (2017).
- [6] Manan, N. S. *et al. J Phys. Chem. B* **115**, 13873-13879 (2011).

2. It was demonstrated that polysulfide becomes colorless after injecting into a carbonate-based electrolyte. However, polysulfide species will react with carbonates. Therefore, believe there experiments shown in Figure 3b will be convoluted by the degree of solubility of the reaction product between carbonates and polysulfide. Overall, please comment on the effect of the polysulfide reaction with carbonate (*J. Phys. Chem. C* 2011, 115, 25132–25137) on Figure 3b.

Reply: Thank you for your valuable comment. As discussed in the above response, the side reaction between Na polysulfides and carbonate solvents is negligible. Therefore, we believe the experiments shown in **Figure 3b** can clearly indicate the formation and diffusion of Na polysulfides.

3. There appears to be a drastic drop in performance when the sulfur areal loading was increased from 1.24 mg cm⁻² to 1.57 mg cm⁻². Please provide an explanation to this phenomenon.

Reply: Thanks for your comments. When the sulfur loading of the electrode was increased to 1.57 mg cm⁻², a capacity decay occurs after about 20 cycles. This is mainly due to the limited viscosity of sodium carboxymethyl cellulose (CMCNa) binder. As seen from **Figure R2**, cracks appear on the surface of S@MPCF electrodes when the electrode thickness was increased from

0.35 mg cm⁻² to 1.57 mg cm⁻² (**Figure R2a-c**), which results in a loss of active materials during cycling and a capacity fading. When styrene butadiene rubber (SBR) was added as the co-binder to improve the viscosity, the surface of S@MPCF electrode exhibits an unbroken morphology without cracks under a sulfur loading of 1.55 mg cm⁻² (**Figure R2d**), and the corresponding capacity retentions of the Na-S cells using CMCNa + SBR binders are significantly improved (**Figure R3**).

Figure R2 | The FE-SEM images of S@MPCF cathode electrodes using **a** CMCNa binder with a sulfur mass loading of 0.35 mg cm⁻²; **b** CMCNa binder with a sulfur mass loading of 1.24 mg cm⁻²; **c** CMCNa binder with a sulfur mass loading of 1.57 mg cm⁻²; **d** CMCNa + SBR binders with a sulfur mass loading of 1.55 mg cm⁻². The cathodes were composed of 80 wt% S@MPCF composite, 10 wt% Super-P and 10 wt% CMCNa in **Figure R2a-c**; 80 wt% S@MPCF composite, 10 wt% Super-P, 4 wt% CMCNa and 6 wt% SBR in **Figure R2d**

Figure R3 | Cycling performances of Na/2 M NaTFSI in PC: FEC (1: 1 by volume) with 10 mM $\text{InI}_3/\text{S}@\text{MPCF}$ cells at 0.1 C using various binders with different sulfur mass loadings.

4. What is the areal sulfur loading for the cells tested in Figure 6 A&B. Please label the areal S loading for both Fig 6 A&B.

Reply: We have followed this suggestion. The sulfur loading in **Figure 6a-b** is $\sim 0.36 \text{ mg cm}^{-2}$. We have labeled it on Page 19 in the revised manuscript.

5. Please show results for sample with just InI_3 added into 1 M NaTFSI-PC electrolyte. This control experiment was not shown in the manuscript.

Reply: Thanks for your valuable comments. The cycling performances of Na/1 M NaTFSI in PC/S@MPCF and Na/1 M NaTFSI in PC with 10 mM $\text{InI}_3/\text{S}@\text{MPCF}$ cells were shown in **Figure S13**. With the addition of InI_3 , the initial discharge capacity increases from 1268 mAh g^{-1} to 1541 mAh g^{-1} , and stabilizes at 481 mAh g^{-1} after 50 cycles at 0.1 C. This verifies that the InI_3 additive can significantly improve the capacity of room-temperature Na-S batteries. We have added this data in the revised **Supporting Information** as the **Figure S13**.

Figure S13 | Cycling performances of Na/S@MPCF cells using 1 M NaTFSI in PC with 10 mM InI₃ and 1 M NaTFSI in PC electrolytes at 0.1 C.

6. The experimental EIS spectrum is not present (or cannot be seen) in Figure S17, please change so it is more apparent.

Reply: Thank you for your comment. We have separated the simulated EIS spectrum and experimental EIS spectrum, and presented as two in **Figure S21**.

Figure S21 | Typical **a** experimental, **b** simulation EIS curves using an equivalent circuit and **c** the simulating result of the Na/2 M NaTFSI in PC: FEC (1: 1 by volume) with 10 mM InI₃/S@MPCF cell.

7. It was mentioned numerous times in the manuscript that an In layer was formed on Na. However, there is not enough evidence to support this claim. Please provide EDS and XPS analysis on the indium and fluorine content on the surface of the sodium metal cycling as shown in Figure 2h. Also, please contrast the morphological difference between the Na metal tested with and without In via a cross-sectional SEM of the Na metal after cycling to support claim of a In layer formation of In.

Reply: We have followed the reviewer's suggestions. Energy dispersive spectroscopy (EDS) and X-ray photoelectron spectroscopy (XPS) measurements were performed to investigate the surface components of Na anodes obtained from Na/1 M NaTFSI-PC/S@MPCF, Na/2 M NaTFSI-PC/S@MPCF, Na/2 M NaTFSI-PC: FEC (1: 1 by volume)/S@MPCF, and Na/2 M NaTFSI-PC: FEC (1: 1 by volume) with 10 mM InI₃/S@MPCF cells. As shown in the XPS spectra in **Figure 3a**, peaks at 686.6 eV and 683.8 eV in fluorine (F) 1s are related to C-F in TFSI⁻ and sodium fluoride (NaF)^[1]. There is a general tendency that the C-F bond in F 1s at about 688 eV^[2] becomes stronger with increasing FEC proportion in electrolytes. The peak intensity of NaF in F 1s also gradually increases with increasing FEC proportion and salt concentration, which verifies the formation of a F-rich SEI layer on the anode surface. This can be further confirmed by the elemental mapping in **Figure S17**. Such F-containing components in the SEI are known to have high mechanical strength (e.g. NaF possesses a shear modulus of 31.4 GPa, more than 10 times higher than that of Na metal^[1]), which enables the SEI layer to suppress the dendritic growth of Na metal. Moreover, the Na anode of the cell using 2 M NaTFSI in PC: FEC with 10 mM InI₃ additive electrolyte exhibits XPS peaks of In 3d at about 457 eV and 445 eV^[3] (**Figure 3a**) with a indium (In) content of 4.8 wt% (**Figure S17**), which indicates the formation of a protective In layer against shuttle effect.

The cross-sectional FE-SEM images of Na anodes obtained from Na/S@MPCF cells using different electrolytes are shown in **Figure S16**. It can be clearly seen from **Figure yc-d** that with the addition of and InI₃ additive, the surfaces of Na anodes become smoother and dendrite growth is dramatically inhibited. We have added above discussion on Page 9 and 11 in the revised manuscript and **Figure S16-17** in the revised **Supporting Information**.

Figure 3 | a XPS spectra of the Na metals from Na/S@MPCF cells using different electrolytes after 50 cycles at 0.1 C.

Figure S17 | EDS mappings of F, S and In elements on the Na anodes obtained from Na/S@MPCF cells using different electrolytes after 50 cycles at 0.1 C, corresponding to **Figure 2 e-h** in the manuscript.

Figure S16 | Cross-sectional FE-SEM images of Na anodes obtained from Na/S@MPCF cells using different electrolytes after 50 cycles at 0.1 C, corresponding to **Figure 2** e-h in the manuscript.

References:

- [1] Seh, Z. W. *et al. ACS Cent. Sci.* **1**, 449-455 (2015).
- [2] Chen, X. *et al. ChemSusChem* **7**, 549-554 (2014).
- [3] Liu, M. *et al. Nano Energy* **40**, 240-247 (2017).

Based on the above comments, the review believes that the manuscript reports good electrochemical performance and fundamental science study, but it's not suitable for publication in Nature Communication due to the insufficient novelty. I recommend submission to a more specialized journal instead.

Reply: Our work provides a fundamental scientific study on the electrolytes for room-temperature Na-S batteries, and successfully achieved outstanding electrochemical performances via electrolyte optimization. We innovatively introduced novel optimization techniques in electrolyte, and also investigated the corresponding functional mechanisms. Most previous works on room-temperature Na-S batteries were concentrated on the modification of electrode materials;^[1] while the effort on electrolyte has rarely been reported^[2]. Therefore, we believe that this work provides a different perspective on the development of Na-S battery system. The electrolyte design strategy in this work can also be extended to other Na-based rechargeable battery system based on a conversion chemistry (e.g. Na-oxygen, Na-selenium, and Na-iodine batteries), Therefore, we think that this work is innovative, which could significantly boost the development of low-cost and high-performance rechargeable batteries.

References:

- [1] Xin, S. *et al. Adv. Mater.* **26**, 1261-1265 (2014); Wang, Y. X. *et al. J. Am. Chem. Soc.* **138**, 16576-16579 (2016); Ye, H. *et al. Proc. Natl. Acad. Sci.* **114**, 13091-13096 (2017); Ma, D. *et al.*

Adv. Func. Mater., **28** 1705537 (2018); Yu, X. *et al. Adv. Energy Mater.*, **5** 1500350 (2015); Carter, R. *et al. Nano Lett.* **17**, 1863-1869 (2017).
[2] Wei, S. *et al. Nat. Commun.* **7**, 11722 (2016); Seh, Z. W. *et al. ACS Cent. Sci.* **1**, 449-455 (2015).

Reviewer #2:

The manuscript described a room-temperature sodium-sulfur battery containing PC/FEC electrolyte with NaTFSi and InI₃. There are a lot of interesting results. It dealt with whole parameters of Na/S cell using sulfur in porous carbon.

1. Sulfur electrode: a. bare sulfur electrode, b. sulfur electrode in multiporous carbon fibers (S@MPCF)

2. Binder: a. CMC, b. PVdF

3. Electrolyte: a. Ether based electrolyte of TEGDME, b. Carbonate based electrolyte: PC, PC-5% FEC, PC-50% FEC – with different concentration of salts NaTFSI and InI₃

Also, there are a lot of interesting results such as optimum electrolyte for long term cycle with high capacity, dissolution behavior of sodium polysulfides in FEC electrolyte, and also optimum binder of CMC instead of PVdF.

However, it is very difficult to understand the main concept and difference from previous works and it is necessary to explain clearly data and discussion.

Reply: Thank you very much for your positive comments on the quality of our paper. Currently the research on room-temperature Na-S battery system is in its infancy, and only few publications about cyclable Na-S batteries have been reported. In this manuscript, we systematically optimized the room-temperature Na-S batteries from several aspects (electrolyte, carbon matrix, binder). The main concept of this work is the innovation of electrolytes. Although ether-based (such as tetraethylene glycol dimethyl ether (TEGDME) and 1,3-dioxolane/1,2-dimethoxyethane (DOL/DME))^[1] and carbonate-based (such as propylene carbonate (PC))^[2] electrolytes have been applied to room-temperature Na-S batteries in previous reports, however, the electrochemical mechanism of different electrolyte solvents on the battery performance is still unclear. In our work, we demonstrate that carbonate-based electrolytes are more suitable than ether-based electrolytes for Na-S battery system due to their high stability to Na metal, limited solubility of Na polysulfides, and low reactivity with Na polysulfides. Based on this, we successfully develop a multifunctional electrolyte containing PC and FEC as co-solvents, highly concentrated Na salt, and InI₃ as an additive. The FEC solvent and high salt concentration not only dramatically reduce the solubility of Na polysulfides, but also construct a robust SEI on the Na anode upon cycling. InI₃ as redox mediator simultaneously increases the kinetic transformation of Na₂S on the cathode and forms a passivating In layer on the anode to prevent it from polysulfide corrosion. This new electrolyte shows excellent electrochemical performances in Na-S cells with various kinds of S@porous carbon composite electrodes (such as S@MPCF and S@CMK-3). Such key findings on electrolyte open up a new direction to boost the performance of room-temperature Na-S batteries.

We also report a novel S@MPCF composite as the cathode material in this work, and firstly discuss the functional mechanism of CMCNa binder on the immobilization of Na polysulfides.

We believe these discoveries could benefit the development of advanced Na-S batteries. To highlight the main concept (electrolyte optimization) of this work, we moved the section of electrode fabrication and the corresponding mechanism discussions to **Supporting Information**.

References:

- [1] Bauer, I. *et al. Chem. Commun.* **50**, 3208-3210 (2014); Kim, I. *et al. J. Electrochem. Soc.* **163**, A611-A616 (2016).
[2] Xin, S. *et al. Adv. Mater.* **26**, 1261-1265 (2014); Wang, Y. X. *et al. J. Am. Chem. Soc.* **138**, 16576-16579 (2016).

Another issue is close correlation between data in manuscript and previous data.

1. The sulfur electrode of S@MPCF has nanosized pores in Fig. S2(e) and 61.09 % sulfur in carbon from TGA of Fig. S2(f). The electrochemical properties critically depend on the position of sulfur whether it is inside nanopore or on the surface. The sulfur inside nanopore can reversibly cycled in carbonate electrolyte but sulfur on the surface shows similar behavior with elemental sulfur, which can only cycle in ether electrolyte such as TEGDME. However, there was no critical explanation about this in the manuscript. In Fig. S4(a) shows irreversible behavior in Li/S cell which is well matched with sulfur on the surface, similar to bare sulfur. Also, this is well matched with Fig. S4(c) in Li/S cell, but it is different from Na/S cell in Fig S4(c) and also other data in Fig.2, which can cycle in carbonate based electrolyte. Since this is very confusing, it should be explained clearly.

Reply: Thanks for your very valuable comments. Li-S batteries can reversibly cycle in carbonate-based electrolytes only when the sulfur is immobilized in the microporous carbon matrixes^[1]. In this case, high-order Li polysulfides do not form and therefore the side reactions between Li polysulfides and carbonate solvents can be avoided. In this work, the average pore size of our carbon matrix, multiporous carbon fibers (MPCF), is about 2.6 nm. Therefore, the side reaction between Li polysulfides and carbonate solvents occurs, and hence the S@MPCF electrode cannot cycles in Li-S cells using PC-based electrolyte, which is similar to the bare sulfur electrode (**Figure S6e**).

However, as shown in **Figure S6**, the S@MPCF electrode can reversibly cycle in Na/S cells using carbonate-based electrolyte (like other S@mesoporous carbon^[2]), while the bare sulfur electrode cannot be cycled. This phenomenon can be explained as follows. As shown in the CV curves in **Figure R4**, a strong peak related to the formation of Na polysulfides appears in the cathodic scan of Na/1 M NaTFSI in PC/bare S cell, corresponding a large initial discharge capacity of 828 mAh g⁻¹ (**Figure S6e**). These are obviously different from the Li/1 M LiTFSI in PC/base S cells with a quite weak reduction peak (**Figure R4**) and a very limited capacity (169 mAh g⁻¹, **Figure S6e**). This illustrates that the side reactions between Na polysulfides and carbonate solvents are less severe than those between Li polysulfides and carbonates, mainly due to the fact that the larger ionic radius of Na⁺ than Li⁺ leads to less dissociation in polar solvents, which results in a lower reactivity of Na⁺-polysulfide⁻ ion pairs than that of Li⁺-polysulfide⁻ ion pairs^[3,4]. However, for the Na/1 M NaTFSI in PC/S cell, no oxidation peak is observed in the subsequent anodic scan (**Figure R4**), meanwhile the discharge capacity is irreversible in the following charging process (**Figure S6e**). This is probably because the volume change (~260 %)

during the cathodic reaction from sulfur to Na_2S is much higher than the volume change from sulfur to Li_2S ($\sim 80\%$)^[5]. As seen from **Figure S6k**, after the initial discharging process, cracks and holes caused by the such huge volume expansion appear on the surface of the bare sulfur electrode, which leads to an irreversible loss of active material and a rapid capacity fading in the following charging. Therefore, it is necessary to immobilize sulfur inside of carbon matrix to relieve the stress caused by the volume change. The S@MPCF electrode, as expected, can maintain a relatively stable capacity in Na/1 M NaTFSI in PC/S@MPCF cells as shown in **Figure S6e** and other data in **Figure S15**. We have added the above explanation in the revised **Supporting Information**.

Figure R4 | The CV curves of Li/1 M LiTFSI in PC/bare S and Na/1 M NaTFSI in PC/bare S cells at a scan rate of 0.1 mV s^{-1} .

Figure S6 | The initial charge/discharge profiles of **f** Li/1 M LiTFSI in PC/bare S and **g** Na/1 M NaTFSI in PC/bare S cells at 0.1 C.

Figure S6 | The FE-SEM images of **i** the initial bare sulfur electrode, the bare sulfur electrodes obtained from **j** Li/1 M LiTFSI-TEGDME/bare S and **k** in Na/1 M NaTFSI-PC/bare S cells after an initial discharging at 0.1 C.

References:

- [1] Xin, S. *et al. J. Am. Chem. Soc.* **134**, 18510-18513 (2012).
- [2] Wang, Y. X. *et al. J. Am. Chem. Soc.* **138**, 16576-16579 (2016).
- [3] Bhattacharyya D. N. *et al. J. Phys. Chem.* **9**, 612-623 (1965).
- [4] Bhattacharyya, D. N. *et al. J. Phys. Chem.* **69**, 608-611 (1965).
- [5] Carter, R. *et al. Nano Lett.* **17**, 1863-1869 (2017).

2. There is important improvement by optimizing electrolyte in Fig. 2 such as high concentration of salts and FEC and InI₃ addition. The electrochemical behavior of basic(or original) sulfur electrode should be confirmed or matched with previous results because there was already good results using sulfur in porous carbon without modification of electrolyte.

Reply: Thank you for your comments. We have followed the reviewer's suggestion and presented the electrochemical behavior of S@MPCF electrode (**Figure S15**). A comparison of previously reported room-temperature Na-S batteries using different S@ porous carbon electrodes are listed in **Table S1**. It is seen that the electrochemical performances of Na-S batteries are greatly affected by the pore size of the carbon host. The electrochemical performance of the S@MPCF (with an average pore size of 2.6 nm) electrode in this work is poorer than those applying S@microporous carbon (with an average pore size of 0.5~1 nm) electrodes, but at the same level as other S@mesoporous carbon electrodes. Furthermore, it is noticed that the cycling capacities of Na-S cells using S@porous carbon electrodes and unmodified electrolytes are generally less than 600 mAh g⁻¹. Only by optimizing the electrolyte (this work) can the cycling capacity of Na-S cells be increased to >900 mAh g⁻¹. We have added this explanation on Page 6~7 in the revised **Supporting Information**.

Figure S15 | Cycling performance of Na/S@MPCF cell with 1 M NaTFSI in PC electrolyte at a current density of 0.1 C.

Table S1 | A comparison of previously reported room-temperature Na-S batteries using different S@ porous carbon electrodes.

Carbon matrix/Average pore size /Sulfur content	Cathode Composition	Electrolyte	Initial Capacity	Cycling Performances	References
Carbon nanotube (CNT)@ microporous carbon (MPC)/~0.5 {Hu, 2017 #69} {Carter, 2017 #19}nm/40 wt%	S/(CNT@MPC): super-P: PVDF=8: 1: 1	1 M NaClO ₄ -PC: EC (v: v=1: 1)	1148 mAh g ⁻¹ at 0.1 C	600 mAh g ⁻¹ at 1 C after 200 cycles	1
Microporous carbon (MC)/<0.7 nm/80 wt%	MC/S composite: super-P: PVDF=7: 2: 1	1 M NaClO ₄ -EC: DEC (v: v=1: 1)	~1600 mAh g ⁻¹ at 0.1 C	392 mAh g ⁻¹ at 1 C after 200 cycles	2
Nanoporous nitrogen doped carbonized ZIF-8 (cZIF-8)/1 nm/50 wt%	cZIF-8/S: Super-P: PVDF=7: 2: 1	1 M NaClO ₄ -TEGDME	873 mAh g ⁻¹ at 0.2 C	500 mAh g ⁻¹ at 0.2 C after 250 cycles	3
Ordered microporous carbon sphere/0.5 nm/35 wt%	S@C: carbon black: CMCNa =8: 1: 1	1 M NaPF ₆ -TEGDME + 0.25 M NaNO ₃	~440 mAh g ⁻¹ at 1C	300 mAh g ⁻¹ at 1 C after 1500 cycles	4
Nitrogen, sulfur-doped hierarchical porous carbon (N, SHPC)/4.2 nm/66.6 wt%	N, SHPC/S: carbon black: (SBR: CMCNa) =7: 2: 1	1 M NaClO ₄ -EC: PC (v: v=1: 1)	455 mAh g ⁻¹ at ~0.07 C	378 mAh g ⁻¹ at ~0.15 C after 350 cycles	5
Interconnected mesoporous carbon hollow nanospheres (iMCHS)/3.6~3.8 nm/46	S@iMCHS: carbon black: CMCNa=7: 1: 2	1.0 M NaClO ₄ -PC: EC (v: v=1: 1) + 5 wt % FEC	1215 mAh g ⁻¹ at ~0.06 C	292 mAh g ⁻¹ at ~0.06 C after 200 cycles	6

wt%					
Multiporous carbon fibers/2.6 nm/60 wt%	S@MPCF: super-P: CMCNa=8: 1: 1	1 M NaTFSI-PC	1292 mA h g ⁻¹ at 0.1 C	286 mA h g ⁻¹ at 0.1 C after 100 cycles	This work

References:

- [1] Xin, S. *et al. Adv. Mater.* **26**, 1261-1265 (2014).
- [2] Hu, L. *et al. ACS Appl. Mater. Interf.* **9**, 13813-13818(2017).
- [3] Chen, Y. -M. *et al. J. Mater. Chem. A* **4**, 12471-12478 (2016).
- [4] Carter, R. *et al. Nano Lett.* **17**, 1863-1869(2017).
- [5] Qiang, Z. *et al. Nano Energy* **32**, 59-66, (2017).
- [6] Wang, Y. X. *et al. J. Am. Chem. Soc.* **138**, 16576-16579 (2016).

Reviewer #3:

The manuscript entitled “A room-temperature sodium-sulfur battery with high capacity and stable cycling performance” reported a carbonate electrolyte with bifunctional additive (InI₃) for sodium-sulfur batteries, in which anion is a redox mediator to improve kinetic of sulfur cathode and cation is able to form robust SEI on anode side. Electrochemical performance of room-temperature Na-S battery improve a lot and the mechanism is discussed with satisfactory characterizations. I would like to recommend its acceptance after my below concern could be addressed.

1. I suggest authors should explain in detail about the cathode. As my understanding, the S@MPCF cathode is a sulfur-PAN composite which can work well in carbonate-based electrolyte because S is completely covalently immobilized in PAN host. The voltage profile shown in Fig. 4d with one plateau around ~1.5 V indicates a solid-solid (S-Na₂S) conversion which is different from the reaction mechanism of that in ether-based electrolyte (solid-intermediate-solid, S-Na₂S_x-Na₂S). That is why this covalent-bonded S composite can work in carbonate-based electrolyte. Generally, based on Li-S chemistry, irreversible reaction can occur between long-chain polysulfides and carbonate electrolytes (ACS Cent. Sci. 2015, 1, 449–455) if S is not covalent-bonded/encapsulated by hosts. Therefore, if the redox reaction in this paper is deemed “solid-solid” reaction, why polysulfides dissolving in the electrolyte/shuttling can occur? Could authors please clarify the difference between Na-S and Li-S chemistry. This is very important to reveal the mechanism why this electrolyte can work well.

Reply: Thank you very much for providing positive comments on our paper and valuable comments. Sulfur-PAN composites were previously reported as cathode material for Li-S^[1] and room-temperature Na-S batteries^[2]. In those works, sulfur and PAN were simply mixed together and heated at ~300 °C under inert atmosphere. During this process, sulfur molecules were covalently attached to the carbon backbone containing pyridinic-N units to form the sulfur-PAN composites. However, in our work, PAN-based nanofibers were totally carbonized at 700 °C to obtain the multiporous carbon fibers (MPCFs), and then sulfur was loaded into these carbon fibers following a melt-diffusion strategy at 155 °C. Although surface bonds between sulfur and the -OH group on the surface of the carbon matrix formed during the preparation process of cathode material, sulfur is obviously not covalently bonded on the PAN host, which is completely different from the previously reported sulfur-PAN composite. Therefore, our cathode material is

sulfur@MPCF, not sulfur-PAN. This can be verified by the XPS result in **Figure S3e**. Furthermore, as seen from the discharge profile in **Figure 4d**, a sloping plateau appears from 1.8 to 1.5 V corresponding to the solid-liquid transition from sulfur to dissolved the Na_2S_x ($x=4\sim 8$), and a long plateau in the range of 1.5 to 1.0 V is related to the formation of Na_2S and Na_2S_2 , which is consistent with the Raman spectra result (**Figure 4a**) and previous reports^[3]. Therefore, we confirm the Na/S@MPCF battery in this work undergoes a “solid-intermediate-solid” reaction, in which soluble polysulfides forms as the intermediate.

Figure S3 | e The S 2p XPS spectra of S@MPCF.

Figure 4 | a Ex-situ Raman spectra of the S@MPCF electrodes obtained from Na/2 M NaTFSI in PC: FEC with 10 mM InI_3 /S@MPCF cells at different charge/discharge potentials; d The 5th charge/discharge profiles of Na/S@MPCF cells using 1 M NaTFSI in PC and 2 M NaTFSI in PC: FEC with InI_3 electrolytes at 0.1 C.

The difference between Li-S and Na-S chemistry can be explained as follow. It is widely recognized that in Li-S batteries, irreversible side reactions can occur between long-chain Li polysulfides and carbonate solvents, which inhibits the solid-intermediate-solid conversion^[4]. However, as seen from **Figure R1**, even though sulfur is not covalent-bonded/encapsulated in hosts and follows a “solid-solid” reaction, the S@MPCF electrodes can maintain a relatively

stable capacity in Na-S cells with PC (propylene carbonate)-based electrolyte. This is mainly because that the larger ionic radius of Na^+ leads to less dissociation in polar solvents, which results in a lower reactivity of Na^+ -polysulfide $^-$ ion pairs than that of Li^+ -polysulfide $^-$ ion pairs^[5]. Hence, the side reactions between Na polysulfides and carbonate solvents are much less than those between Li polysulfides and carbonates, which enables the application of carbonate electrolytes in the Na-S cells based on “solid-intermediate-solid” chemistry.

Figure R1 | The cycling performances of Na-S and Li-S batteries using S@MPCF cathode in PC-based electrolytes at 0.1 C.

References:

- [1] Wang, J. *et al. Adv. Mater.* **14**, 13-14 (2002); Wei, S. *et al. J. Am. Chem. Soc.* **137**, 12143-12152 (2015).
- [2] Hwang, T. H. *et al. Nano Lett.* **13**, 4532-4538 (2013).
- [3] Wang, Y. X. *et al. J. Am. Chem. Soc.* **138**, 16576-16579 (2016); Manthiram, A. *et al. Small* **11**, 2108 (2015).
- [4] Gao, J. *et al. J. Phys. Chem. C* **115**, 25132-25137, (2011).
- [5] Bhattacharyya D. N. *et al. J. Phys. Chem.* **9**, 612-623 (1965); Bhattacharyya, D. N. *et al. J. Phys. Chem.* **69**, 608-611 (1965).

2. As previous work, microporous porous only allow small sulfur molecular parasitic which also guarantee that sulfur is able to work in non-polysulfide dissolve carbonate electrolyte. I am curious about sulfur structure in MPCF with microporous (<1 nm) and surprise the high sulfur loading in MPCF (60 wt%) which is higher than most of microporous carbon (<40 wt% sulfur loading). Could the author explain its unique?

Reply: Thanks for your valuable comment. The high sulfur loading of MPCF is attributed to its hierarchical structure. As seen from the TEM image in **Figure S2b** and the pore size distribution in **Figure S2e** inset, massive mesoporous (with a pore size of about 5~10 nm) originated from the ferric acetylacetonate (FeAcAc) template appears in the MPCFs; meanwhile many micropores

originated from the alkali activation process homogeneously distribute in the outer layer of MPCFs. The pore volume of MPCF is as high as $1.6 \text{ cm}^3 \text{ g}^{-1}$ and the mean pore size is around 2.6 nm. Considering the density in molten sulfur is $\sim 1.82 \text{ g cm}^{-3}$ and the density of sulfur powder is $1.96\sim 2.07 \text{ g cm}^{-3}$ [1], the theoretical sulfur loading in MPCF is calculated to be 74.4~76.8 wt%, which is well consistent with our experimental data. We have added the above discussion on Page 5~6 in **Supporting Information**.

Figure S2 | b The TEM image of the MPCFs.

Figure S2 | e (inset) The pore size distribution of MPCFs.

Reference:

[1] Ji, X. *et al. Nat. Mater.* **8**, 500-506 (2009).

3. To further clarify the influence of concentration, I suggest authors provide the voltage profiles and cycling performance using 1M NaTFSI-PC and 2M NaTFSI-PC for comparison.

Reply: Thanks for your value suggestion. We have added the voltage profiles and cycling performances of Na/1 M NaTFSI in PC/S@MPCF and Na/2 M NaTFSI in PC/S@MPCF cells for comparison. When the salt concentration of the electrolyte is increased from 1 M to 2 M, the reversible capacity of the Na/2 M NaTFSI in PC/S@MPCF cell significantly improves from 314 mAh g⁻¹ to 589 mAh g⁻¹ after 50 cycles at 0.1 C. Please refer to the added **Figure S9** in the revised **Supporting Information**.

Figure S9 | The 1st, 5th and 50th charge/discharge profiles of Na/S@MPCF cells using **a** 1 M NaTFSI in PC and **b** 2 M NaTFSI in PC electrolytes at 0.1 C; **c** The cycling performances of Na/S@MPCF cells using different electrolytes at 0.1 C.

4. I notice that the electrolyte/sulfur ratio is 20 $\mu\text{L}/\text{mg}$. I am wondering if the parameter is optimized or not? What is the common value of this ratio in most references for Na-S batteries?

Reply: Thanks for your important comments.

The value of electrolyte/sulfur ratio in this work has been optimized. Too much electrolyte can lead to an aggravated dissolution of the Na polysulfides and the shuttle effect; while too little electrolyte will cause a difficult infiltration to the active materials. As seen from **Figure R5**, the cell with 20 $\mu\text{L mg}^{-1}$ electrolyte showed the best cycling performances (1120 mAh g⁻¹ after 50 cycles at 0.1 C). As seen from the **Table R1**, the common value of electrolyte/sulfur ratio for room-temperature Na-S batteries is 12 $\mu\text{L mg}^{-1}$ ~24 $\mu\text{L mg}^{-1}$ based on previous reports, which is well consistent with our work.

Figure R5 | Cycling performances of Na/2 M NaTFSI in FEC with InI₃/S@MPCF cells with different electrolyte/sulfur ratios at 0.1 C.

Table R1 Comparison of the electrolyte/sulfur ratios in the previously reported room-temperature Na-S batteries.

Cathode activated materials	Electrolyte	Electrolyte/sulfur ratio (μL mg ⁻¹)	Cycling performances	References
Na ₂ S/activated carbon nanofiber	1.5 M NaClO ₄ -TEGDME + 0.2 M NaNO ₃	24	550 mAh g ⁻¹ at 0.2 C after 100 cycles	1
Freestanding carbon fiber cloth/sulfur composites	1.5 M NaClO ₄ -TEGDME + 0.2 M NaNO ₃	12	120 mAh g ⁻¹ at 0.1 C after 300 cycles	2
Sulfur/carbon nanofibers	1 M NaPF ₆ -TEGDME	~25	~550 mAh g ⁻¹ at 0.1 C after 20 cycles	3
Na ₂ S-multi-walled carbon nanotube fabric electrodes	1.5 M NaClO ₄ -TEGDME + 0.3 M NaNO ₃	~15	560 mAh g ⁻¹ at 0.1 C after 50 cycles	4
S@multiporous carbon fibers	2 M NaTFSI-PC: FEC (v: v=1: 1) + 10 mM InI₃	20	927 mAh g⁻¹ at 0.1 C after 200 cycles	This work

References:

- [1]Yu, X. *et al. Chem. Mater.* **28**, 896-905, (2016).
- [2]Lu, Q. *et al. Energy Storage Mater.* **8**, 77-84 (2017).
- [3]Seh, Z. W., *et al. ACS Cent. Sci.* **1**, 449-455 (2015).

[4]Yu, X. *Chemistry* **21**, 4233-4237 (2015).

5. Gravimetric energy density of Na-S batteries in Figure 6 e is overestimated that would mislead the reader. The overall capacity of Sulfur-Carbon composition should be used instead of pure sulfur. The 1477 Wh/kg is only based on the sulfur weight, but when turning to gravimetric energy density, it should be $1477 * 60\%$ (sulfur content) = 886 Wh/kg, which is comparable with the current lithium-ion cathodes.

Reply: Thanks for your comment. We have revised the data presented in **Figure 6c**. It can be seen that the gravimetric energy density of our Na-S battery based on the mass of S@C composite, ~886 Wh kg⁻¹, is still much higher than most Li-ion and Na batteries.

Figure 6 | e Comparison of practical specific capacities and energy densities of Na batteries and Li batteries with representative reported cathode materials. The specific capacities and energy densities are evaluated by the mass of cathodic active material only.

6. I suggest the author supply the first charge-discharge profile which is very important for the reader to get the full picture of Na-Sulfur batteries. And I disagree the author's explanation the huge irreversible capacity loss in the first cycle and low coulombic efficiency " The initial irreversible conversion of Na₂S together with the formation of SEI on the anode result in an initial Coulombic efficiency of 79.1 % at 0.1 C, 71.9 % at 0.5 C and 61.6 % at 1 C (the Coulombic efficiency is calculated as percentage of the charge capacity in respect to the discharge capacity).-----" In the work, the anode is excessive, so SEI formation can not influence efficiency in half cell and the efficiency only express the information about cathode, so I think the irreversible capacity loss should result from cathode. Please give the reasonable explanation?

Reply: Thanks for your valuable suggestion. We have added the first charge-discharge profiles of Na/1 M NaTFSI in PC/S@MPCF and Na/2 M NaTFSI in PC: FEC (1: 1 by volume) with 10 mM InI₃/S@MPCF cells at 0.1 C. Please see **Figure S23** in the revised **Supporting Information**.

We appreciate the reviewer's comment, and agree with that the SEI formation on the anode cannot influence the efficiency in half cell due to the excess of anode, and the efficiency only expresses the irreversible conversion of Na_2S . We have accordingly revised the "The initial irreversible conversion of Na_2S together with the formation of SEI on the anode result in an initial Coulombic efficiency..." as "The initial irreversible conversion of Na_2S results in an initial Coulombic efficiency..." on Page 17 in the revised manuscript.

Figure S23 | The charge/discharge profiles of Na/S@MPCF cells using **a** 1 M NaTFSI in PC and **b** 2 M NaTFSI in PC: FEC (1: 1 by volume) with 10 mM InI₃ electrolytes at different current densities.

Reviewers' comments:

Reviewer #1 (Remarks to the Author):

The effort spent on the revision is acknowledged. Most of the technical problems have been answered. Unfortunately, most of the ideas presented here are still not new. The authors did try to address this overarching problem in writing. However, I do not believe the reasons provided by the authors are sufficient:

1) FEC is not just used as a common alloy-anode stabilizer but also well known in Li metal anode systems. (Journal of The Electrochemical Society, 160 (10) A1894-A1901 (2013)). I believe the fact that the observed low binding energy between the SEI and polysulfide is somewhat interesting, it does not warrant publication in Nat. Comm.

2) It can be agreed that the Nano Energy 30, 825–830 (2016) does not discuss about the polysulfide solubility. However, high salt concentration is well known to suppress the polysulfide dissolution through the common-ion effect. Chem. Commun., 2013, 49, 2004-2006. This point cannot be taken as novelty

3) From the rebuttal letter "In our work, InI3 additive was applied to overcome the intrinsic issue of Na-S cells, which is the poor transformation kinetics related to the conversion from Na₂S₂ or Na₂S to long-chain polysulfides due to dramatic volume change caused by the large ion size of Na⁺." This sentence indicates that the main novelty of their use of InI₃ is to assist to enhance the poor kinetics of Na₂S charging. However, this is well-known in the very similar Li₂S cathode systems, where InI₃ has been used to reduce the 1st charging overpotential of Li₂S (Nano Energy 40 (2017) 240–247). From the arguments stated, InI₃ does not play a different role in Li₂S cathode system when compared to the Na-S systems. It acts as both a redox mediator to relieve slow kinetics in addition to prevent polysulfide corrosion at the anode's surface.

While it does take effort to combine different techniques from different fields, each technique is well known enough that such combination-type work might be considered rather trivial. I would say the single most interesting component of this paper is the data that indicates a stable Na polysulfide formation in carbonate electrolyte (while Li polysulfide is well known to be not). This could be impactful for future applications. However, even with this new data, we do not believe it is suitable for publication in Nat. Comm. as the underlying key-points of the paper is inevitably just a combination of previous works.

Reviewer #2 (Remarks to the Author):

The manuscript was fully revised about my comment and concern and added many data. In my opinion, it is acceptable for the publication of Nature Communications. For clear understanding, I suggest the following comments.

1. It is necessary to describe in detail about the preparation method for SEM of Fig. S16 and Fig. 2 because sodium is very reactive in air atmosphere.
2. It is necessary to describe the sample of Fig. S2 because the loading of sulfur (0.35 – 4.64mg/cm³) should affect the composition of sulfur on the surface or pore which can show the different TGA curve.

Reviewer #3 (Remarks to the Author):

The cover letter responses my concern carefully and also supply relative experiment to support their

explanation. I feel well for the reversed manuscript and would like to recommend its acceptance at current stage.

Reviewer #4 (Remarks to the Author):

Electrolytes are often the key component responsible for poor cell performances, and critical innovation are needed if better performances are expected from new battery chemistries. The interactions (and the in situ formed passivation layers) between the electrolyte and the electrodes will define the electrochemical performance of the batteries. The present work aims to address the poor cycling performance of Na-S battery systems on the basis of the "cocktail optimized" electrolyte approach. First of all, there are some interesting results on the designing of the electrolyte for the Na-S battery. However several critical issues, which might mislead the researchers, should be carefully addressed before this paper being accepted by Nature Commun.

The major issues are as follows:

1) For the synthesis of the S@MPCF cathode material, the authors first heated the S and MPCF mixture to a temperature of 155 °C. And then further heated at 300 °C for 1 h in Ar. Why the authors further heated the composite to the high temperature of 300 °C in Ar? However, in Figure S2f, TGA curves of S@MPCF under N₂ clearly show that almost all of the S will be evaporated before the temperature is raised to 300 °C. The authors should carefully address these self-contradictory results. Considering the TGA curves are obtained in the ramping program with a certain heating rate, if the S@MPCF was kept at 300 °C in inert gas for a while, all of the S should disappear due to the volatilization.

2) I totally agree with the 1st comment of the Reviewer #3, and the reply or the evidences are not so solid. As my understanding, as raising the temperature to 300 °C or even higher, the vapor pressure of the S is very high, and some S₄ or S₂ molecules will be in the vapor. These forms of the Sulfur will more easily interact/react with defected carbon or PAN, or be trapped by the micro-pores in the carbon. These types of carbon/small-S composite usually have a good cycling performance in carbonate electrolyte. What is the form of S trapped in the MPCF? Only the XPS characterization is not so solid. In the reply letter, the authors claimed that in the discharge profile (Figure 4d) there are two plateaus and a sloping plateau from 1.8 V to 1.5 V corresponding to the solid-liquid transition from sulfur to dissolved the Na₂S_x (x=4-8). However, in the small-sulfur/microporous-C composite (Guo et al, A high-energy room-temperature sodium-sulfur battery, Adv. Mater., 26, 1261-1265.) almost the same discharge curves were observed. So if just considering the discharge curve, the S/MPCF composite is more like the small-sulfur composite, not the traditional S composite.

3) To avoid the above misunderstanding, it is highly suggested that the authors use the traditional method to prepare the S/MPCF composite (155 °C), rather than raised the temperature to 300 °C.

4) As stated in the manuscript, some Na polysulfides as intermediates will form and result in shuttle reactions. Due to the shuttle reactions, the charging capacities should be higher than the discharge capacities, leading to the CE > 100% (Here, the Coulombic efficiency is calculated as percentage of the charge capacity in respect to the discharge capacity). The Li-S batteries are the good examples. However, why all of the CEs are lower than 100% (as shown in Figure 2c)?

5) To demonstrate the self-discharge processes, the authors added some sulfur powder and Na metal foil into the different electrolyte. However, since the authors utilized a special synthesis method with a higher preparing temperature of 300 °C, it is highly suggested that the authors could directly utilize the S@MPCF electrode not S powder to simulate the self-discharge processes.

Other minor issues:

6) What is the heating rate for the TGA test?

7) In the experimental section, line 461, what is MPSFs? Is it typo?

8) Several highly reversible Na-S batteries have been reported, (such as Ref. 24 in the manuscript, A high-energy room-temperature sodium-sulfur battery, Adv. Mater., 26, 1261-1265.) and also get a

highly reversible Na-S battery with a reversible capacity of > 1000 mAh/g. Therefore it is suggested to delete or revise the statement of "To the best of our knowledge, this is the best electrochemical performance achieved for room-temperature Na-S batteries."

9) It is highly suggested to narrow the axis range of the Coulombic efficiency, such as to 60-105. Besides, in line 373, it is suggested to revise the phrase of "high Coulombic efficiency ($\sim 100\%$)". For the Coulombic efficiency, 99% or 99.9% will result in totally different cycling performance in the full battery.

10) In Fig 4b: the authors should plot the charge/discharge as a loop if they want to highlight the hysteresis.

Response to reviewers' comments

Reviewer #1:

The effort spent on the revision is acknowledged. Most of the technical problems have been answered. Unfortunately, most of the ideas presented here are still not new. The authors did try to address this overarching problem in writing. However, I do not believe the reasons provided by the authors are sufficient:

1) FEC is not just used as a common alloy-anode stabilizer but also well known in Li metal anode systems. (Journal of The Electrochemical Society, 160 (10) A1894-A1901 (2013)). I believe the fact that the observed low binding energy between the SEI and polysulfide is somewhat interesting, it does not warrant publication in Nat. Comm.

2) It can be agreed that the Nano Energy 30, 825–830 (2016) does not discuss about the polysulfide solubility. However, high salt concentration is well known to suppress the polysulfide dissolution through the common-ion effect. Chem. Commun., 2013, 49, 2004-2006. This point cannot be taken as novelty.

3) From the rebuttal letter “In our work, InI₃ additive was applied to overcome the intrinsic issue of Na-S cells, which is the poor transformation kinetics related to the conversion from Na₂S₂ or Na₂S to long-chain polysulfides due to dramatic volume change caused by the large ion size of Na⁺.” This sentence indicates that the main novelty of their use of InI₃ is to assist to enhance the poor kinetics of Na₂S charging. However, this is well-known in the very similar Li₂S cathode systems, where InI₃ has been used to reduce the 1st charging overpotential of Li₂S (Nano Energy 40 (2017) 240–247). From the arguments stated, InI₃ does not play a different role in Li₂S cathode system when compared to the Na-S systems. It acts as both a redox mediator to relieve slow kinetics in addition to prevent polysulfide corrosion at the anode’s surface.

While it does take effort to combine different techniques from different fields, each technique is well known enough that such combination-type work might be considered rather trivial. I would say the single most interesting component of this paper is the data that indicates a stable Na polysulfide formation in carbonate electrolyte (while Li polysulfide is well known to be not). This could be impactful for future applications. However, even with this new data, we do not believe it is suitable for publication in Nat. Comm. as the underlying key-points of the paper is inevitably just a combination of previous works.

Reply: Thanks for the reviewer's comments. In this work, we not only reveal a stable Na polysulfide formation in carbonate electrolyte, but also provide multiple optimizations of electrolyte for Na-S batteries, which are essentially different from previous reported techniques. The point-to-point explanations are listed below to clearly elucidate the innovations presented in this paper.

1) In the work reported in *J. Electrochem. Soc.* **160**, A1894-A1901 (2013), FEC was employed as an electrolyte solvent to stabilize the Li anodes. **In our work, FEC was used as a co-solvent for room-temperature Na-S batteries. It not only benefits the formation of a stable F-rich SEI film on the Na metal surface upon cycling, but also possesses a low binding energy with Na polysulfides which successfully enables polysulfides to remain in the cathode instead of dissolving into electrolyte.** Obviously, there is no similarity between our work and previous reports.

2) In the work reported in *Chem. Commun.* **49**, 2004-2006 (2013), 5.0 M LiTFSI in the co-solvent of dimethoxyethane (DME) and 1, 3-dioxolane (DOL) was applied to suppress the polysulfide dissolution in Li-S batteries. In our work, we successfully proved that 2 M NaTFSI concentration in a carbonate-based electrolyte not only effectively reduced the solubility of the Na polysulfides, but also formed a stable F-rich SEI and a dendrite-free Na surface during cycling. **We also demonstrate that the capacity of Na-S batteries gradually decreases with excessively high salt concentrations (beyond 2 M) owing to the high electrolyte viscosity.** This clearly shows that the high concentrated salt in electrolytes for Na-S batteries follows an obviously different mechanism compared with that for Li-S batteries.

3) In the work reported in *Nano Energy* **40**, 240-247 (2017), InI₃ was proposed for SnO₂/Li₂S full batteries. The deposited In layer protected the Li anode from side reactions, and the I⁻/I³⁻ redox mediator promoted the activation of Li₂S cathode. In Na-S batteries, the transformation kinetics related to the conversion from Na₂S to long-chain polysulfides is much poorer than that from Li₂S to long-chain polysulfides, mainly due to dramatic volume change caused by the larger ion size of Na⁺. In our work, for the first time, we reveal that the low Coulombic efficiency and rapid capacity fading of S-based cathodes in Na-S batteries caused by such poor kinetics can be greatly overcome by redox mediators such as I⁻/I³⁻. The InI₃ can also form a protective In layer on the Na anode

against polysulfide corrosion. Therefore, the InI_3 plays a more essential role in Na-S batteries compared with that in Li-S batteries.

Based on above discussions, it can be clearly seen that this work is highly innovative. The research on room-temperature Na-S batteries system is currently still in its infancy and only few reports about cyclable Na-S batteries have been published. We believe that this research is a major scientific breakthrough and of great significance to the development of room-temperature Na-S batteries.

Reviewer #2:

The manuscript was fully revised about my comment and concern and added many data. In my opinion, it is acceptable for the publication of Nature Communications. For clear understanding, I suggest the following comments.

1. It is necessary to describe in detail about the preparation method for SEM of Fig. S16 and Fig. 2 because sodium is very reactive in air atmosphere.

Reply: Thanks for your valuable comment.

In **Figure S16** and **Figure 2**, the Na anode samples for SEM characterization were prepared by disassembling the cycled Na-S cells, and repeatedly rinsed the obtained anodes with DME and vacuum dried at 50 °C for 1 h to remove the residual solvent. All above procedures were operated in a glove box in which the moisture and oxygen are controlled below 0.5 ppm. The air-sensitive Na anode samples were then rapidly transferred into the vacuum chambers of SEM under the protection of vacuum box. We have added above explanation in the revised manuscript. Please refer to the page 22 in the revised manuscript.

2. It is necessary to describe the sample of Fig. S2 because the loading of sulfur ($0.35 - 4.64\text{mg}/\text{cm}^3$) should affect the composition of sulfur on the surface or pore which can show the different TGA curve.

Reply: Thanks for your comment. In **Figure S2f**, the sulfur loading in the S@MPCF sample is ~61.09 wt %. We have added this on Page 6 in the revised Supporting Information.

Reviewer #3:

The cover letter responds my concern carefully and also supply relative experiment to support their explanation. I feel well for the reversed manuscript and would like to recommend its acceptance at current stage.

Reply: Thank you very much for your positive comments on the high quality of our revised manuscript .

Reviewer #4:

Electrolytes are often the key component responsible for poor cell performances, and critical innovation are needed if better performances are expected from new battery chemistries. The interactions (and the in situ formed passivation layers) between the electrolyte and the electrodes will define the electrochemical performance of the batteries. The present work aims to address the poor cycling performance of Na-S battery systems on the basis of the “cocktail optimized” electrolyte approach. First of all, there are some interesting results on the designing of the electrolyte for the Na-S battery. However several critical issues, which might mislead the researchers, should be carefully addressed before this paper being accepted by Nature Commun. The major issues are as follows:

1) For the synthesis of the S@MPCF cathode material, the authors first heated the S and MPCF mixture to a temperature of 155 °C. And then further heated at 300 °C for 1 h in Ar. Why the authors further heated the composite to the high temperature of 300 °C in Ar? However, in Figure S2f, TGA curves of S@MPCF under N₂ clearly show that almost all of the S will be evaporated before the temperature is raised to 300 °C. The authors should carefully address these self-contradictory results. Considering the TGA curves are obtained in the ramping program with a certain heating rate, if the S@MPCF was kept at 300 °C in inert gas for a while, all of the S should disappear due to the volatilization.

Reply: Thanks for your important comments.

In this work, the temperature of the sulfur-impregnation process was further increased to 300 °C with the purpose of infiltrating the sulfur on the surface of MPCF into the pores. Previously, Guo, J. *et al.* has compared the effect of heating temperature (160 °C and 300 °C) on the distribution of sulfur in the porous carbon. They found that after a sulfur-impregnation process at 160 °C, sulfur

was mainly located in relatively large-size voids or on the surface of the porous carbon. When the temperature increased to 300 °C, a large portion of sulfur successfully diffused into the smaller voids of the porous carbon. As a result, the utilization of sulfur has been improved^[1]. Therefore, a further heating at 300 °C for sulfur-impregnation was applied in this work.

Furthermore, as commented by the reviewer, the S@MPCF sample was exposed in a N₂ flow during the TGA test. Therefore, sulfur evaporated rapidly at 300 °C. However, during the sulfur-impregnation process, the sulfur/MPCF mixture was put in a sealed container. Consequently, the evaporation of sulfur at high temperature (155 °C and 300 °C) is dramatically suppressed and negligible, which is consistent with previous reports^[2]. We have added this explanation on Page 21 in the revised manuscript, and on Page 6 in the revised Supporting Information.

References:

[1] Guo, J. *et al. Nano Lett.* **11**, 4288-4294 (2011).

[2] Zhen, L. *et al. Nature Comm.* **6**, 8850 (2015).

2) I totally agree with the 1st comment of the Reviewer #3, and the reply or the evidences are not so solid. As my understanding, as raising the temperature to 300 °C or even higher, the vapor pressure of the S is very high, and some S₄ or S₂ molecules will be in the vapor. These forms of the Sulfur will more easily interact/react with defected carbon or PAN, or be trapped by the micro-pores in the carbon. These types of carbon/small-S composite usually have a good cycling performance in carbonate electrolyte. What is the form of S trapped in the MPCF? Only the XPS characterization is not so solid. In the reply letter, the authors claimed that in the discharge profile (Figure 4d) there are two plateaus and a sloping plateau from 1.8 V to 1.5 V corresponding to the solid-liquid transition from sulfur to dissolved the Na₂S_x (x=4-8). However, in the small-sulfur/microporous-C composite (Guo et al, A high-energy room-temperature sodium-sulfur battery, Adv. Mater., 26, 1261-1265.) almost the same discharge curves were observed. So if just considering the discharge curve, the S/MPCF composite is more like the small-sulfur composite, not the traditional S composite.

Reply: Thanks for your comment.

At a heating temperature of 300 °C, some S₄ or S₂ molecules in in the vapor will be trapped by the micro-pores in the carbon. However, it should be noticed that during the subsequent cooling process,

these short-chain sulfur molecules recover to S_8 in the carbon pores. Only when the pore size in the porous carbon is less than 0.5 nm, these short-chain sulfur molecules trapped inside will not re-form to S_8 due to the confined effect^[1]. In our work, the average pore size of MPCF is ~ 2.6 nm. Therefore, sulfur trapped in the MPCF is theoretically in form of S_8 instead of short-chain sulfur molecules, which is well consistent with the results of Raman spectra (**Figure R1**).

Furthermore, we have compared the discharge-charge curves of Na-S cells applying the S@MPCF and the small-sulfur/microporous carbon composite (S/(CNT@MPC))^[2] as cathode materials, respectively. It is seen from **Figure S23a** that for the Na/S@MPCF cell, a sloping plateau from 2.1 to 1.5 V and a long plateau in the range of 1.5 to 1.0 V appear in the initial discharge process, corresponding to the solid-liquid transition from sulfur to dissolved Na_2S_x ($x=4\sim 8$) and the formation of Na_2S and Na_2S_2 , respectively. These two plateaus show a capacity ratio of about 1: 3, which is well consistent with the theoretical value. Such discharge-charge behavior is consistent with other reported Na-S batteries using sulfur@mesoporous carbon cathodes^[3]. In sharp contrast, the Na-S cell with the S/(CNT@MPC) cathode shows two sloped discharge plateaus (divided by a dashed line at ~ 1.4 V) with almost equal capacity contributions in the first cycle (**Figure R2**), and the discharge products of such two plateaus are Na_2S_2 and Na_2S , respectively^[2]. Therefore, the discharge behavior of S@MPCF in Na-S batteries is same as that of the sulfur@mesoporous carbon composites, but obviously different from that of the small-sulfur/microporous carbon composites.

Figure R1 | Raman spectra of the S@MPCF and pure sulfur powder.

Figure S23 | The charge/discharge profiles of Na/S@MPCF batteries using 1 M NaTFSI in PC electrolyte at different current densities.

Figure R2 | The charge/discharge profiles of the S/(CNT@MPC) cathode in Na-S cells at 0.1 C^[2].

Reference:

- [1] Xin, S. *et al.* *J. Am. Chem. Soc.*, **134**, 18510-18513 (2012).
- [2] Xin, S. *et al.* *Adv. Mater.*, **26**, 1261-1265 (2014).
- [3] Wang, Y. X. *et al.* *J. Am. Chem. Soc.*, **138**, 16576-16579 (2016).

3) To avoid the above misunderstanding, it is highly suggested that the authors use the traditional method to prepare the S/MPCF composite (155 °C), rather than raised the temperature to 300 °C.

Reply: Thanks for your suggestion.

We have followed the reviewer's suggestion and tested the cycling performance of the Na/S@MPCF cell using S@MPCF obtained by heating the sulfur/MPCF mixture at 155 °C (labeled as "S@MPCF-155 °C"). Compared with the S@MPCF further heated at 300 °C (labeled as "S@MPCF-300 °C"), the S@MPCF-155 °C cathode shows a slightly lower initial Coulombic efficiency (63.5 % vs. 79.1 %) and reversible capacity after 20 cycles (1064 mA h g⁻¹ vs. 1155.1 mA h g⁻¹). We have added this result as **Figure S2g** in the revised Supporting Information.

Figure S2 | g Cycling performances of Na/2 M NaTFSI in PC: FEC (1: 1 by volume) with 10 mM InI₃/S@MPCF cells at 0.1 C, using S@MPCF powders obtained by heating the sulfur/MPCF mixture at 155 °C with or without a further heating treatment at 300 °C.

4) As stated in the manuscript, some Na polysulfides as intermediates will form and result in shuttle reactions. Due to the shuttle reactions, the charging capacities should be higher than the discharge capacities, leading to the CE > 100% (Here, the Coulombic efficiency is calculated as percentage of the charge capacity in respect to the discharge capacity). The Li-S batteries are the good examples. However, why all of the CEs are lower than 100% (as shown in Figure 2c)?

Reply: Thanks for your very significant comments.

As stated by the reviewer, in Li-S batteries the Coulombic efficiency is sometimes higher than

100.0 % due to the shuttle of polysulfides. It is noticed that this phenomenon also exists in Na-S batteries. We have narrowed the axis range of the Coulombic efficiency in **Figure 2c**. It is clearly seen that the Coulomb efficiencies of Na/2 M NaTFSI in PC: FEC/S@MPCF cell in some cycles are higher than 100.0 % (*e. g.* 101.1% in the 14th cycle and 100.5% in the 39th cycle) at 0.1 C. We have revised the **Figure 2c** in Page 10 in the revised manuscript accordingly.

Figure 2 | c Cycling performances of Na/S@MPCF cells using electrolytes with various concentration of NaTFSI in PC: FEC (1: 1 by volume) solvents at 0.1 C.

5) To demonstrate the self-discharge processes, the authors added some sulfur powder and Na metal foil into the different electrolyte. However, since the authors utilized a special synthesis method with a higher preparing temperature of 300 °C, it is highly suggested that the authors could directly utilize the S@MPCF electrode not S powder to simulate the self-discharge processes.

Reply: Thanks for your significant comments.

We have followed the reviewer's suggestion and performed a self-discharge experiment using S@MPCF electrodes instead of sulfur powder. As shown in **Figure S18c**, the electrolyte 1# (1 M NaTFSI in PC) and 3# (2 M NaTFSI in PC) become darker in color after aging at 60 °C for 36 h, demonstrating that the sulfur powder continuously dissolves in the simplistic electrolyte and electrochemically reacts with Na metal to form highly soluble polysulfides with dark colors. However, the 2# (1 M NaTFSI in PC: FEC) and 4# (2 M NaTFSI in PC: FEC) electrolytes maintain nearly transparent after aging, which could be ascribed to the low solubility of Na polysulfides in FEC solvent or high-salt concentration solution. The 5# electrolyte, 2 M NaTFSI in PC: FEC (1: 1 by volume) with 10 mM InI₃, maintains the pristine yellow color from the InI₃ additive during the

aging test. These phenomena are basically consistent with the previous self-discharge experiment using sulfur powder (**Figure 3b**). However, the changes in color are less obvious due to the confinement of sulfur in the MPCF porous carbon. We have added this result as **Figure S18c** in the revised Supporting Information.

Figure S18 / c Visual observation of Na polysulfides formation in five electrolyte samples (1#: 1 M NaTFSI in PC; 2#: 1 M NaTFSI in PC: FEC (1: 1 by volume); 3#: 2 M NaTFSI in PC; 4#: 2 M NaTFSI in PC: FEC (1: 1 by volume); 5#: 2 M NaTFSI in PC: FEC (1: 1 by volume) with 10 mM InI_3) along with aging time at 60 °C. The S@MPCF electrodes with sulfur loading of $\sim 1.2 \text{ mg cm}^{-2}$ and Na metal foils were added into the electrolytes to simulate the self-discharge processes.

Other minor issues:

6) *What is the heating rate for the TGA test?*

Reply: The heating rate of TGA test is $10 \text{ }^\circ\text{C min}^{-1}$. We have added this information on Page 6 in the revised Supporting Information.

7) *In the experimental section, line 461, what is MPSFs? Is it typo?*

Reply: Thanks for your comment and sorry for our carelessness. We have revised “MPSFs” as “MPCFs” on Page 21 in the manuscript.

8) *Several highly reversible Na-S batteries have been reported, (such as Ref. 24 in the manuscript, A high-energy room-temperature sodium-sulfur battery, Adv. Mater., 26,1261-1265.) and also get a highly reversible Na-S battery with a reversible capacity of > 1000 mAh/g. Therefore it is suggested to delete or revise the statement of “To the best of our knowledge, this is the best electrochemical performance achieved for room-temperature Na-S batteries.”*

Reply: Thank you for the precious advice. We have revised “to the best of our knowledge, this is the best electrochemical performance achieved for room-temperature Na-S batteries” as “to the best of our knowledge, this is **one of** the best electrochemical performances achieved for room-temperature Na-S batteries” in the revised manuscript.

9) *It is highly suggested to narrow the axis range of the Coulombic efficiency, such as to 60-105. Besides, in line 373, it is suggested to revise the phrase of “high Coulombic efficiency (~100%)”. For the Coulombic efficiency, 99% or 99.9% will result in totally different cycling performance in the full battery.*

Reply: We have narrowed the Coulombic efficiency axis ranges of **Figure 2c**, **Figure 6a**, **Figure S3a**, **Figure S22** and **Figure S24a** in the revised manuscript and Supporting Information. In addition, we have revised the phrase of “high Coulombic efficiency (~100%)” as “high Coulombic efficiency of **98.5 %~100.4 %**” in Page 17 in the revised manuscript.

10) *In Fig 4b: the authors should plot the charge/discharge as a loop if they want to highlight the hysteresis.*

Reply: Thank you for your valuable suggestion. We have re-plotted the charge/discharge curves as a loop in **Figure 4d** in the revised manuscript.

Figure 4 | d The 5th charge/discharge profiles of Na/S@MPCF cells using 1 M NaTFSI in PC and 2 M NaTFSI in PC: FEC (1: 1 by volume) with InI₃ electrolytes at 0.1 C.

Reviewers' comments:

Reviewer #4 (Remarks to the Author):

Some of the technical questions were answered. However, there are still several issues need to be addressed:

- 1) About the first question. In the original version of the paper, they did not mention the sulfur/MPCF mixture was put in a sealed container. In the experimental section, they only said "MPSFs and nano sulfur powder (Dk Nano technology, Beijing) were ground together at a weight ratio of 4:6, and subsequently heated at 155 °C for 10 h and further heated at 300 °C for 1 h in Ar." This is a key experimental procedure, which might mislead the researchers. The authors should carefully check other experimental procedures to ensure the results can be repeated by other researchers.
- 2) About the second question. The authors claimed that the average pore size of MPCF is 2.6 nm based on BET. However, in the Figure S2e (inset), it seems that the volumes of smaller pores (< 2 nm) are much more than 2.6 nm pores. If the micro-sized pores dominate the pores, small-sulfur might be trapped in these micro-sized pores?
- 3) The authors have fully revised the third comment. This is good.
- 4) About the fourth question. The reply is totally wrong. If the shuttle reaction dominates the side reactions during cycling. The average CEs (Here, the Coulombic efficiency is calculated as percentage of the charge capacity in respect to the discharge capacity) should be a little higher than 100%. It can be seen that all of the average CEs for the four samples are obviously lower than 100%. Here, I mean the average CEs, not some cycle's CE. Only one or two cycle's CE higher than 100% are not due to shuttle reactions. If the shuttle reaction does not dominate the side reactions. Lots of content in the paper should be revised, because the authors discussed a lot of "shuttle reactions" in the paper.
- 5) About the fifth question. Compared with the Figure 3b, it seems that the color change is not so distinct. It seems the colors of 2# and 4# also became a little darker? It seems there are some shallow on the samples when taking the photo for the "aging for 36 h" samples?

Response to reviewer's comments

Reviewer #4:

Some of the technical questions were answered. However, there are still several issues need to be addressed:

1) About the first question. In the original version of the paper, they did not mention the sulfur/MPCF mixture was put in a sealed container. In the experimental section, they only said "MPSFs and nano sulfur powder (Dk Nano technology, Beijing) were ground together at a weight ratio of 4:6, and subsequently heated at 155 °C for 10 h and further heated at 300 °C for 1 h in Ar." This is a key experimental procedure, which might mislead the researchers. The authors should carefully check other experimental procedures to ensure the results can be repeated by other researchers.

Reply: Thanks for your comment and sorry for our carelessness. We have followed the reviewer's comment and accordingly added the detailed experimental procedures and highlighted them on Page 21 in the revised manuscript and Page 2 in the revised Supporting Information to ensure the accuracy.

2) About the second question. The authors claimed that the average pore size of MPCF is 2.6 nm based on BET. However, in the Figure S2e (inset), it seems that the volumes of smaller pores (< 2 nm) are much more than 2.6 nm pores. If the micro-sized pores dominate the pores, small-sulfur might be trapped in these micro-sized pores?

Reply: As stated by the reviewer, the MPCFs contain micropores with sizes of 0.5~2 nm (**Figure R1**). However, it should be emphasized that the short-chain sulfur molecules would be trapped in the porous carbon only when the carbon pore size is less than 0.5 nm^[1]. Therefore, sulfur trapped in the MPCF is in form of S₈ instead of short-chain sulfur molecules since the pore size distribution of MPCF does not meet such requirement. Moreover, the Na/S@MPCF cell shows a similar charge-discharge behavior (**Figure S23a**) as the previously reported Na-S cells using sulfur@mesoporous carbon cathodes^[2], which further verifies that the sulfur in the MPCFs is not in the form of small-sulfur.

Figure R1 | The pore size distributions of MPCF and S@MPCF powder.

References

- [1] Xin, S. *et al. J. Am. Chem. Soc.*, **134**, 18510-18513 (2012).
 [2] Wang, Y. X. *et al. J. Am. Chem. Soc.*, **138**, 16576-16579 (2016).

3) *The authors have fully revised the third comment. This is good.*

Reply: Thanks for your positive comment on our revision.

4) *About the fourth question. The reply is totally wrong. If the shuttle reaction dominates the side reactions during cycling. The average CEs (Here, the Coulombic efficiency is calculated as percentage of the charge capacity in respect to the discharge capacity) should be a little higher than 100%. It can be seen that all of the average CEs for the four samples are obviously lower than 100%. Here, I mean the average CEs, not some cycle's CE. Only one or two cycle's CE higher than 100% are not due to shuttle reactions. If the shuttle reaction does not dominate the side reactions. Lots of content in the paper should be revised, because the authors discussed a lot of "shuttle reactions" in the paper.*

Reply: Thanks for your comment.

As stated by the reviewer, the average Coulombic efficiencies (calculated as percentage of the charge capacity in respect to the discharge capacity) of the Na-S cells in this work are less than 100%. However, in this manuscript, we have clearly claimed that the Coulombic efficiencies of Na-S batteries are affected by multiple factors rather than simply dominated by the shuttle effect. As stated on Page 4 in the revised manuscript, the transition from

short-chain Na polysulfides or Na₂S to long-chain polysulfides is kinetically poor, which greatly reduces the Coulombic efficiency during the cycling. The continuous damage/regeneration of the SEI on the anode also decreases the Coulombic efficiency of Na-S cells. As a result, an average Coulombic efficiency less than 100 % was observed in these Na-S batteries. After the electrolyte optimization, the Coulombic efficiency of Na-S batteries has been significantly improved to ~100 % by enhancing the transformation kinetics of Na₂S in the cathode and suppressing the Na dendrite growth (**Figure 2c** and **d**), which is well consistent with above explanation. We have highlighted above discussion about Coulombic efficiency on Page 4 in the revised manuscript.

5) About the fifth question. Compared with the Figure 3b, it seems that the color change is not so distinct. It seems the colors of 2# and 4# also became a little darker? It seems there are some shallow on the samples when taking the photo for the “aging for 36 h” samples?

Reply: Thanks for your valuable comment.

The color change of the self-discharge experiment using S@MPCF was not so distinct. Although the colors of sample 2# and 4# slightly became darker, such color changes are much less obvious than the samples 1# and 3#. To further discuss above phenomenon, self-discharge experiment using sulfur powder as an alternative and the corresponding UV-vis spectra have been shown in **Figure 3b** and **Figure S18**, which are well consistent with this result.

REVIEWERS' COMMENTS:

Reviewer #4 (Remarks to the Author):

The authors have replied the comments, and this paper can be accepted.

One minor issue is about the Coulombic efficiency (CE). The CE that the authors actually tested is the S/C composite's CE not the full cell's CE because of the extremely large excess of Na metal compared with the capacity of S/C cathode. In this cell, S/C composite is the working electrode (cathode), and Na metal is the counter and reference electrode (anode). The CE that the authors tested is the working electrode's CE. Therefore, the continuous damage/regeneration of the SEI on the anode will not affect the tested CE. However, in the reply of the comment #4, the authors stated "The continuous damage/regeneration of the SEI on the anode also decreases the Coulombic efficiency of Na-S cells". This statement is wrong.

Reviewer #4:

The authors have replied the comments, and this paper can be accepted.

One minor issue is about the Coulombic efficiency (CE). The CE that the authors actually tested is the S/C composite's CE not the full cell's CE because of the extremely large excess of Na metal compared with the capacity of S/C cathode. In this cell, S/C composite is the working electrode (cathode), and Na metal is the counter and reference electrode (anode). The CE that the authors tested is the working electrode's CE. Therefore, the continuous damage/regeneration of the SEI on the anode will not affect the tested CE. However, in the reply of the comment #4, the authors stated "The continuous damage/regeneration of the SEI on the anode also decreases the Coulombic efficiency of Na-S cells". This statement is wrong.

Reply: We appreciate the reviewer's comment, and agree with that the damage/regeneration of SEI on the Na anode cannot influence the Coulombic efficiency of Na-S half cells due to the excess of anode. Therefore, the Coulombic efficiency only reflects the irreversible conversion of Na₂S as well as the shuttle effect. Thanks for this valuable comment.